# Decreased microRNA levels lead to deleterious increases in neuronal M2 muscarinic receptors in Spinal Muscular Atrophy models

Patrick J O'Hern[1], Inês do Carmo G. Gonçalves[2], Johanna Brecht[2], Eduardo Javier López Soto[1], Jonah Simon[1], Natalie Chapkis[1], Diane Lipscombe[1,3], Min Jeong Kye[2], Anne C Hart[1]*

[1]Department of Neuroscience, Brown University, Providence, United States; [2]Institute of Human Genetics, University of Cologne, Cologne, Germany; [3]Brown Institute for Brain Science, Providence, United States

**Abstract** Spinal Muscular Atrophy (SMA) is caused by diminished Survival of Motor Neuron (SMN) protein, leading to neuromuscular junction (NMJ) dysfunction and spinal motor neuron (MN) loss. Here, we report that reduced SMN function impacts the action of a pertinent microRNA and its mRNA target in MNs. Loss of the *C. elegans* SMN ortholog, SMN-1, causes NMJ defects. We found that increased levels of the *C. elegans* Gemin3 ortholog, MEL-46, ameliorates these defects. Increased MEL-46 levels also restored perturbed microRNA (miR-2) function in *smn-1(lf)* animals. We determined that miR-2 regulates expression of the *C. elegans* M2 muscarinic receptor (m2R) ortholog, GAR-2. GAR-2 loss ameliorated *smn-1(lf)* and *mel-46(lf)* synaptic defects. In an SMA mouse model, m2R levels were increased and pharmacological inhibition of m2R rescued MN process defects. Collectively, these results suggest decreased SMN leads to defective microRNA function *via* MEL-46 misregulation, followed by increased m2R expression, and neuronal dysfunction in SMA.

*For correspondence:
anne_hart@brown.edu

**Competing interests:** The authors declare that no competing interests exist.

## Introduction

Spinal Muscular Atrophy (SMA) is an autosomal recessive neurodegenerative disease and the leading genetic cause of infant death in the US (*Cusin et al., 2003*; *Pearn, 1978*). SMA is caused by homozygous deletion or mutation of the *SMN1 (Survival Motor Neuron 1)* gene, resulting in reduced Survival of Motor Neuron (SMN) protein levels (*Lefebvre et al., 1995*). SMN expression is ubiquitous, but particularly essential for motor neuron survival (*Lefebvre et al., 1997*). Disease severity, as well as spinal cord α-MN dysfunction and degeneration, correlates with the extent of SMN loss (*Lefebvre et al., 1997*). Understanding why SMN loss impairs function should offer insight into SMA and may reveal therapeutic targets. SMN is conserved across species (*Miguel-Aliaga et al., 1999*). Studies of various SMA models suggest a role for SMN in several cellular processes including snRNP assembly (*Golembe et al., 2005*; *Yong et al., 2002*), messenger RNA (mRNA) transport (*Fallini et al., 2011*), and local translation (*Dimitriadi et al., 2010*; *Kye et al., 2014*). SMN function, however, has not been linked definitively to MN degeneration or synaptic transmission defects caused by SMN loss.

microRNAs (miRNAs) are non-coding RNAs that often repress protein translation, by a mechanism that requires miRNA binding to the 3'UTR of mRNA targets. Disruption of the miRNA pathway in spinal MNs leads to severe degeneration (*Haramati et al., 2010*). SMN loss alters levels and/or

**eLife digest** Spinal muscular atrophy is a genetic disease that causes muscles to gradually weaken. In people with the disease, the nerve cells that control the movement of muscles – called motor neurons – deteriorate over time, hindering the person's mobility and shortening their life expectancy. Spinal muscular atrophy is usually caused by genetic faults affecting a protein called SMN (which is short for "Survival of motor neuron") and recent research suggested that disrupting this protein alters the function of short pieces of genetic material called microRNAs. However, the precise role that microRNAs play in the disease and their connection to the SMN protein was not clear.

MicroRNAs interfere with the production of proteins by disrupting molecules called messenger RNAs, which are temporary strings of genetic code that carry the instructions for making protein. By disrupting messenger RNAs, microRNAs can delay or halt the production of specific proteins. This is an important part of the normal behavior of a cell, but disturbing the activity of microRNAs can lead to an unwanted rise or fall in crucial proteins.

O'Hern et al. made use of engineered nematode worms and mice that share genetic features with spinal muscular atrophy patients, including disruption of the gene responsible for producing the SMN protein. These animal models of the disease were used to examine the relationship between decreased SMN levels and microRNAs in motor neurons. The experiments showed that reduced SMN activity affects a specific microRNA, which in turn causes motor neurons to produce more of a protein called m2R. This protein is a receptor for a molecule, called acetylcholine, which motor neurons use to send signals to muscle cells.

Increased m2R may be detrimental to motor neurons. As such, O'Hern et al. decreased m2R protein activity to determine whether this could reverse the defects in motor neurons that arise in the animal models of the disease. Indeed, blocking this receptor rescued some of the defects seen in the animal models, supporting the link to spinal muscular atrophy.

Several treatments that block m2R are already available to treat other conditions. As such, the next step is to determine whether these existing treatments are able to protect mice models of spinal muscular atrophy against muscle deterioration or increase their lifespan. If successful, this could open new avenues for the development of treatments in people.

activity of specific miRNAs (*Haramati et al., 2010*; *Kye et al., 2014*; *Valsecchi et al., 2015*; *Wang et al., 2014*), but the cellular mechanisms leading to altered miRNA expression and/or function are unknown. The RNA helicase Gemin3 associates with both SMN and RNA-induced silencing complex components (*Charroux et al., 1999*; *Höck et al., 2007*; *Hutvágner and Zamore, 2002*; *Meister et al., 2005*; *Mourelatos et al., 2002*; *Murashov et al., 2007*). Gemin3 and SMN levels decrease concomitantly, suggestive of a functional link (*Feng et al., 2005*; *Helmken et al., 2003*).

We took advantage of the *C. elegans* SMA model to examine the connection between SMN, Gemin3, and miRNA function. *SMN1*, *Gemin3,* and multiple miRNA pathway components are conserved in *C. elegans* (*Grishok et al., 2001*; *Miguel-Aliaga et al., 1999*; *Minasaki et al., 2009*). Loss-of-function (lf) mutations in *smn-1*, the *C. elegans* ortholog of *SMN1*, cause behavioral and morphological abnormalities, premature death, and sterility (*Briese et al., 2009*; *Sleigh et al., 2011*). *smn-1 (lf)* animals also have neuromuscular junction (NMJ) defects, suggesting a functional role for SMN-1 in MNs (*Briese et al., 2009*). MNs in *smn-1(lf)* animals do not die, likely because of their short lifespan. However, *smn-1(lf)* neuromuscular defects may correspond to the early stages of SMA pathogenesis, characterized by NMJ dysfunction prior to MN degeneration (*Miguel-Aliaga et al., 1999*; *Yoshida et al., 2015*). We find that the *C. elegans* Gemin3 ortholog, MEL-46, is perturbed by SMN-1 loss, impacting miR-2 suppression of the M2 muscarinic receptor ortholog, GAR-2 (*Lee et al., 2000*). Across species in SMA mouse models, we find decreased levels of miR-128, a potential miR-2 ortholog, and increased expression of the GAR-2 ortholog, m2R. Notably, m2R inhibition ameliorates axon outgrowth defects in MNs from a SMA mouse model, consistent with our results in *C. elegans*.

## Results

### MEL-46 (Gemin3) is required for NMJ function

The *C. elegans* Gemin3 ortholog is MEL-46. Homozygous loss of *smn-1* or *mel-46* results in lethality (*Briese et al., 2009*; *Miguel-Aliaga et al., 1999*), but maternal loading of *smn-1* or *mel-46* mRNA and protein allows many homozygous, loss of function animals to survive into the last larval stage, called L4 (*Miguel-Aliaga et al., 1999*; *Minasaki et al., 2009*). Loss of *smn-1* results in neuromuscular defects including decreased pharyngeal pumping rates, followed by overtly altered locomotion and subsequent death (*Briese et al., 2009*). Like *smn-1* loss in L4 stage animals, we found that *mel-46 (tm1739)* homozygous loss of function animals had severely decreased pharyngeal pumping rates. Pharyngeal pumping was restored to normal rates in *mel-46(tm1739)* animals using a previously described, broadly expressed *mel-46* rescue array (also referred to as [*mel-46(+)*#1]), which utilizes the *mel-46* promoter (*Figure 1A*) (*Minasaki et al., 2009*). *mel-46* partial loss of function alleles, *yt5* and *ok3760*, also caused pumping defects as did global *mel-46* RNA interference (*RNAi*) or cholinergic neuron-specific *mel-46(RNAi)* (*Figure 1—figure supplement 1A–E*). We conclude that MEL-46 is necessary for normal neuromuscular function.

SMN-1 is required for normal NMJ function in *C. elegans* cholinergic MNs (*Dimitriadi et al., 2016*). Aldicarb is an acetylcholinesterase inhibitor that leads to acetylcholine accumulation in the NMJ and consequently, paralysis (*Mahoney et al., 2006*). The time course of aldicarb-induced paralysis was slowed by decreased SMN-1 activity (*Dimitriadi et al., 2016*). We tested if a decrease in MEL-46 function causes similar resistance to aldicarb and found that *mel-46* loss of function resulted in aldicarb resistance across multiple alleles (*Figure 1B*; *Figure 1—figure supplement 1F and G*), reminiscent of *smn-1* loss. Reintroduction of *mel-46* using the [*mel-46(+)*#1] rescue array restored aldicarb sensitivity in *mel-46(tm1739)* animals. Tissue-specific knock-down of *mel-46* in cholinergic neurons resulted in aldicarb resistance, thus confirming that MEL-46 function is required in cholinergic neurons, as is SMN-1 (*Figure 1C*) (*Dimitriadi et al., 2016*). We also showed that knock-down of *mel-46* or *smn-1* in inhibitory GABAergic neurons resulted in aldicarb hypersensitivity (*Figure 1—figure supplement 1H*). Our findings, taken together with previous work, suggest that MEL-46 and SMN-1 are required in both cholinergic and GABAergic neurons for normal NMJ function.

*smn-1* loss causes changes in presynaptic protein localization (*Dimitriadi et al., 2016*). Do similar changes occur in *mel-46(tm1739)* animals? We evaluated localization of presynaptic proteins SNB-1 (synaptobrevin) and APT-4 (AP2 α-adaptin) in cholinergic dorsal A-type (DA) MNs of *mel-46(tm1739)* animals (*Ch'ng et al., 2008*; *Sieburth et al., 2005*). SNB-1 is a v-SNARE protein required for SV exocytosis, while APT-4 associates with clathrin-coated endocytic vesicles (*Kamikura and Cooper, 2006*; *Nonet et al., 1998*). In the dorsal cord, cholinergic DA MNs do not have presynaptic inputs; they form *en passant* presynaptic connections in a punctate pattern (*Ch'ng et al., 2008*; *White et al., 1976*). Three parameters were measured to evaluate fluorescently labeled SNB-1 and APT-4 localization to presumptive synapses: puncta width (μm), intensity (AU), and linear density (puncta/μm), as previously described (*Kim et al., 2008*). Loss of *smn-1* or *mel-46* resulted in similar SNB-1 synaptic localization defects: decreased SNB-1 puncta width, intensity and linear density (*Figure 1D–G*) (*Dimitriadi et al., 2016*). Loss of *smn-1* leads to decreased APT-4 puncta width and intensity, but increased linear density (*Dimitriadi et al., 2016*). *mel-46(tm1739)* animals also had increased APT-4 linear density, but no changes in puncta width or intensity compared to controls (*Figure 1—figure supplement 2A–C*). Therefore, decreased *mel-46* causes synaptic protein defects that overlap partially with defects observed when SMN-1 levels decrease. Given the similarities between SMN-1 and MEL-46 loss in aldicarb resistance, decreased pharyngeal pumping rates, and defective synaptic protein localization, we decided to explore whether SMN-1 and MEL-46 act in common pathways required for NMJ function.

### Perturbed MEL-46 (Gemin3) function likely contributes to synaptic defects in *smn-1(lf)* animals

MEL-46 might act together with or downstream of SMN-1 in pathways necessary for NMJ function. To test these and other possibilities, we generated integrated multicopy transgenic lines expressing GFP-tagged MEL-46 expressed under control of the *unc-17* cholinergic-specific promoter (*Figure 2A*). MEL-46::GFP was found in both the cell bodies and processes of neurons. No obvious

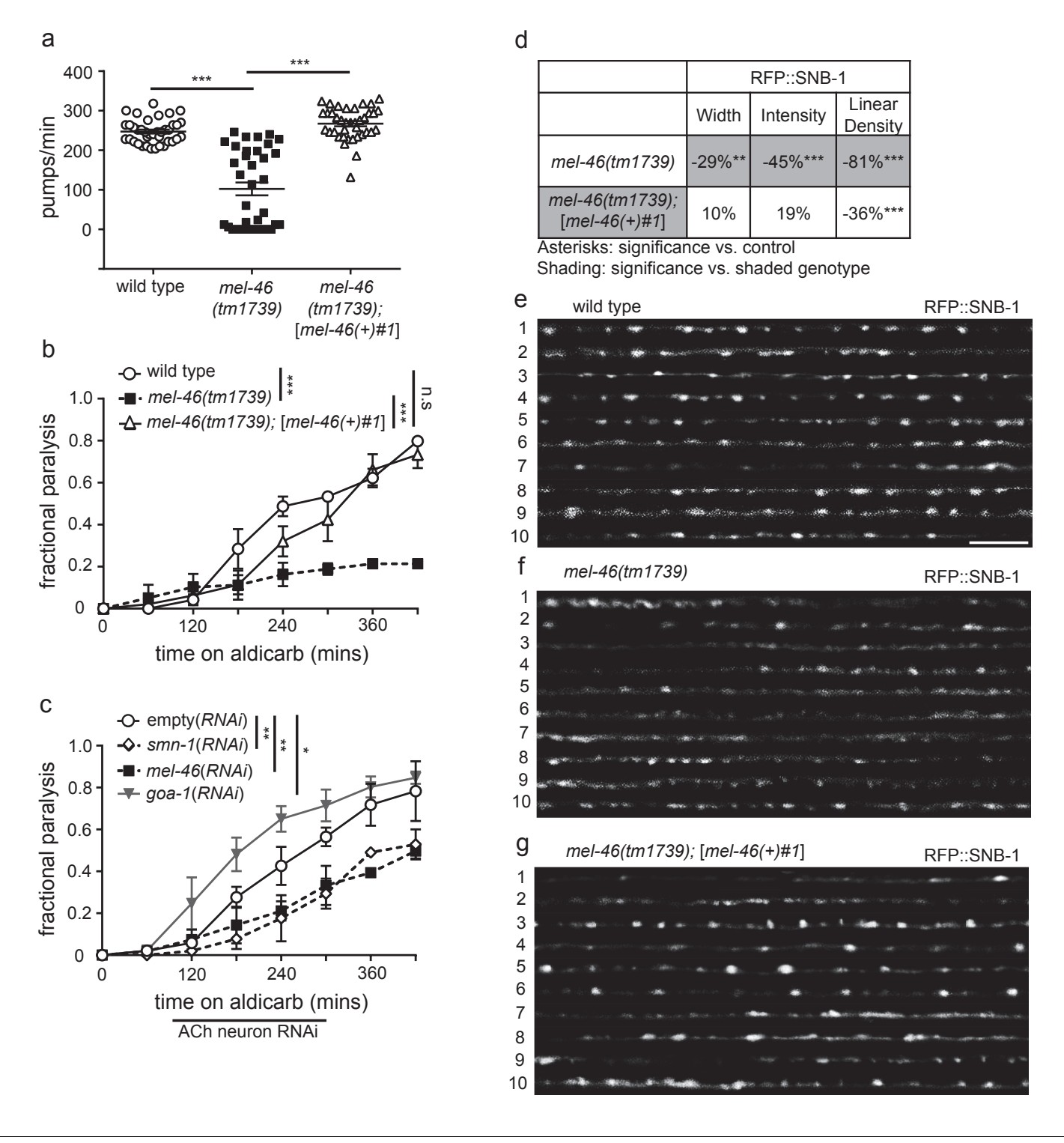

**Figure 1.** Decreased MEL-46 function in *C. elegans* results in defective NMJ signaling. (a) *mel-46(tm1739)* animals had reduced pharyngeal pumping rates *versus* wild type (N2) control animals. Defects were fully rescued by global expression of MEL-46 behind its own promoter ([*mel-46(+)#1*]). Mean ± SEM; Mann-Whitney *U*-test, two-tailed. (b) *mel-46(tm1739)* animals paralyzed more slowly when exposed to aldicarb, an acetylcholinesterase inhibitor. Time course for paralysis on 1 mM aldicarb for wild type (N2), *mel-46(tm1739)*, and *mel-46(tm1739);[mel-46(+)#1]* early larval stage L4 animals. Reintroduction of *mel-46* restored normal aldicarb sensitivity. Log-rank test. (c) Cholinergic neuron-specific *mel-46(RNAi)* causes resistance to aldicarb. Time course for paralysis on 1.5 mM aldicarb for empty(*RNAi*), smn-1(*RNAi*), mel-46(*RNAi*), and goa-1(*RNAi*) young adult animals. Animals sensitive to RNAi in only cholinergic neurons (XE1581) were fed bacteria expressing double-stranded RNA (dsRNA) against *mel-46*, *smn-1*, or *goa-1* (positive
*Figure 1 continued on next page*

*Figure 1 continued*

control). Control animals were fed bacteria expressing an empty vector control: empty(*RNAi*). Data set previously published without *mel-46(RNAi)* (*Dimitriadi et al., 2016*). Log-rank test. (d) *mel-46(tm1739)* animals had reduced RFP::SNB-1 (synaptobrevin). Percent change from wild type (N2) control for RFP::SNB-1 in the dorsal cord of *mel-46(tm1739)* and *mel-46(tm1739);[mel-46(+)#1]* animals for 'punctaanalyzer' parameters: puncta width (μm), intensity (AU), and linear density (number/μm). Asterisks denote significance compared to wild type; shading indicates significant change for *mel-46(tm1739)* versus *mel-46(tm1739);[mel-46(+)#1]*. Mann-Whitney *U*-test, two-tailed. Expression of *mel-46* rescued RFP::SNB-1 puncta width defects in *mel-46(tm1739)* animals (wild type *versus mel-46(tm1739);[mel-46(+)#1]* $p=0.82$; *mel-46(tm1739)* versus *mel-46(tm1739);[mel-46(+)#1]* $p=0.03$), rescued SNB-1 puncta intensity defects (wild type *versus mel-46(tm1739);[mel-46(+)#1]* $p=0.85$; *mel-46(tm1739)* versus *mel-46(tm1739);[mel-46(+)#1]* $p=0.005$) and partially ameliorated SNB-1 puncta linear density defects (wild type *versus mel-46(tm1739);[mel-46(+)#1]* $p=0.0004$; *mel-46(tm1739)* versus *mel-46 (tm1739);[mel-46(+)#1]* $p=0.0001$). (h–j) Representative images of RFP::SNB-1 expressed in the dorsal cord of cholinergic DA MNs for wild type, *mel-46 (tm1739)*, and *mel-46(tm1739);[mel-46(+)#1]* animals. These images were taken as part of data collection. Scale bar, 5 μm. For statistical analyses in all figures: *$p\leq0.05$, **$p<0.01$, ***$p<0.001$. A helpful summary of *C. elegans* phenotypes reported throughout this article for selected loss of function alleles can be found in *Supplementary file 1*. Additional information on *C. elegans* strains used in *Figures 1–6* is provided in *Supplementary file 2A*.

The following source data and figure supplements are available for figure 1:

**Source data 1.** Raw Data for *Figure 1—figure supplement 2*.
**Figure supplement 1.** MEL-46(Gemin3) is necessary for proper NMJ function.
**Figure supplement 1—source data 1.** Raw Data for *Figure 1—figure supplement 1*.
**Figure supplement 2.** MEL-46(Gemin3) loss causes increased APT-4(AP2 α-adaptin) linear density.
**Figure supplement 2—source data 1.** Raw Data for *Figure 1—figure supplement 2*.

changes were seen in cytoplasmic MEL-46::GFP, leading us to evaluate localization of MEL-46::GFP in MN dorsal cord processes in *smn-1(ok355)* animals. Because *ok355* deletion in *smn-1* leads to a complete loss of function, *smn-1(ok355)* animals were maintained over an *hT2* balancer and sterile *smn-1(ok355)* homozygous progeny carry some maternally-loaded SMN-1 protein (*Briese et al., 2009*). We found that MEL-46::GFP localizes to small granular structures in dorsal cord processes in control (*smn-1(+)*) and *smn-1(ok355)* animals. Our finding is consistent with previous work showing that Gemin3 localizes to granular structures in mammalian neurites; Gemin3 co-localizes with SMN in 50–60% of these granules, along with multiple mRNAs (*Todd et al., 2010a, 2010b*; *Zhang et al., 2006*). In *smn-1(ok355)* animals, we found that the density of MEL-46::GFP-positive granular structures was doubled compared to *smn-1(+)* controls (*Figure 2B–D*). Furthermore, the mean intensity of MEL-46::GFP fluorescence and the maximum fluorescence for each sample were decreased in *smn-1(ok355)* animals (*Figure 2B–D*; *Figure 2—figure supplement 1A*). These results suggest that decreased SMN-1 leads to MEL-46 mislocalization in cholinergic MN processes and diminished MEL-46 levels in granules. Our findings suggest that SMN-1 impairs MEL-46 function, which could contribute to *smn-1(ok355)* synaptic defects (*Dimitriadi et al., 2016*). To test this hypothesis, we increased *mel-46* gene dosage in *smn-1(ok355)* animals using the [*mel-46(+)#1*] rescue array and showed that this ameliorated *smn-1(ok355)* aldicarb resistance defects (*Figure 2E*). We also showed that increasing *mel-46* specifically in cholinergic neurons, using the cholinergic-specific *unc-17* (ACh) promoter in an integrated array, referred to as [ACh::*mel-46*::GFP], rescued *smn-1(ok355)* aldicarb resistance (*Figure 2—figure supplement 1B and C*). The aldicarb resistance observed with broad expression of *mel-46* in control animals (*Figure 2E*) was not observed when we overexpressed *mel-46* in cholinergic neurons only. Under this condition we observed mild hypersensitivity in one integrated line (referred to as [ACh::*mel-46*::GFP#1]) (*Figure 2—figure supplement 1C*) and no difference from control animals in a second line (referred to as [ACh::*mel-46*::GFP#2]) (*Figure 2—figure supplement 1B*). It is possible that high levels of MEL-46 in cholinergic neurons cause aldicarb hypersensitivity, whereas broad overexpression of MEL-46 may impact NMJ function independent of cholinergic neurons. Taken together, our results suggest that loss of SMN-1 negatively impacts MEL-46 function, resulting in perturbed NMJ signaling. Our finding is consistent with observations in humans that reduced human SMN levels result in Gemin3 downregulation (*Feng et al., 2005*; *Helmken et al., 2003*),

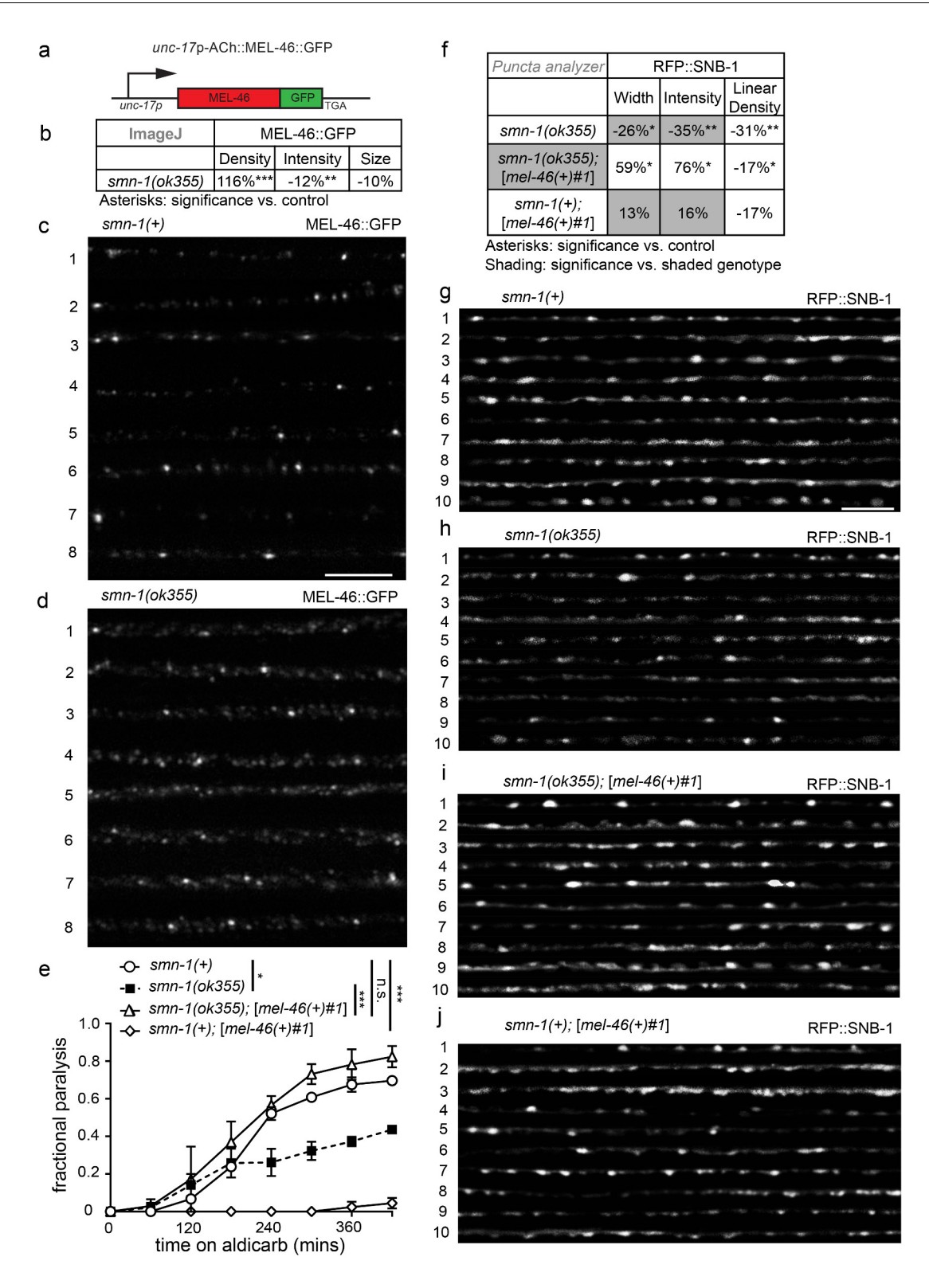

**Figure 2.** MEL-46 localization and levels are perturbed in *smn-1(lf)* animals. (a) Illustration: *mel-46* was tagged with GFP at the C-terminus and expression was driven by the cholinergic (ACh) *unc-17* promoter. Two lines were generated by UV integration. (b) *smn-1(ok355)* animals exhibited mislocalization and reduction of MEL-46::GFP in dorsal cord processes of cholinergic neurons. MEL-46::GFP localizes to granular punctate structures in dorsal cord processes. Percent change from *smn-1(+)* control for MEL-46::GFP in the dorsal cord of *smn-1(ok355)* animals for ImageJ parameters:

*Figure 2 continued*

puncta density (puncta/area), puncta intensity (AU), and puncta size (pixels/puncta). The ImageJ analysis was used instead of the 'punctaanalyzer' program since MEL-46::GFP had a scattered non-linear pattern in *smn-1(ok355)* animals; a linear pattern is necessary for accurate 'punctaanalyzer' analysis. Asterisks denote significance compared to wild type. Mann-Whitney *U*-test, two-tailed. (**c–d**) Representative images of MEL-46::GFP in dorsal cord cholinergic DA MN processes for control *smn-1(+)* and *smn-1(ok355)* animals. These images were taken as part of data collection. Scale bar, 5 μm. (**e**) Increasing expression of *mel-46* rescued *smn-1(ok355)* aldicarb response defects. Time course for paralysis on 1 mM aldicarb for *smn-1(+)*, *smn-1(ok355)*, *smn-1(ok355);[mel-46(+)#1]*, and *smn-1(+);[mel-46(+)#1]* early larval stage L4 animals. *smn-1(+);[mel-46(+)#1]* animals were resistant to paralysis by aldicarb. Log-rank test. (**f**) Increasing *mel-46* rescued *smn-1(ok355)* RFP::SNB-1 synaptic localization defects. Percent change from *smn-1(+)* control for RFP::SNB-1 in the dorsal cord of *smn-1(ok355), smn-1(ok355);[mel-46(+)#1]*, and *smn-1(+);[mel-46(+)#1]* animals for 'punctaanalyzer' parameters: puncta width (μm), intensity (AU), and linear density (number/μm). Asterisks denote significance compared to *smn-1(+)* control; shading indicates significant difference from *smn-1(ok355);[mel-46(+)#1]*. Mann-Whitney *U*-test, two-tailed. Expression of *mel-46* restored RFP::SNB-1 puncta width defects (*smn-1(+)* control *versus smn-1(ok355);[mel-46(+)#1]* p=0.05; *smn-1(ok355) versus smn-1(ok355);[mel-46(+)#1]* p=0.001), rescued SNB-1 puncta intensity defects (*smn-1(+)* control *versus smn-1(ok355);[mel-46(+)#1]* p=0.035; *smn-1(ok355) versus smn-1(ok355);[mel-46(+)#1]* p=0.0004), but did not rescue SNB-1 puncta linear density defects (*smn-1(+)* control *versus smn-1(ok355);[mel-46(+)#1]* p=0.036; *smn-1(ok355) versus smn-1(ok355);[mel-46(+)#1]* p=0.19). (**G–J**) Representative images of cholinergic DA MN RFP::SNB-1 in the dorsal nerve cord of *smn-1(+), smn-1(ok355)* and *smn-1(ok355);[mel-46(+)#1]*. These images were taken as part of data collection. Scale bar, 5 μm.

The following source data and figure supplements are available for figure 2:

**Source data 1.** Raw Data for *Figure 2*.
**Figure supplement 1.** Expressing *mel-46* restores neuronal defects in *smn-1(lf)* animals.
**Figure Supplement 1—source data 1.** Raw Data for *Figure 2—figure supplement 1*.
**Figure supplement 2.** Expressing *mel-46* does not increase SMN-1 levels.
**Figure Supplement 2—source data 1.** Raw Data for *Figure 2—figure supplement 2*.

The interdependency we report, between SMN-1 and MEL-46 levels, may be specific to particular tissues and/or neural circuits. For example, pharyngeal pumping rate defects were not ameliorated in *smn-1(ok355)* animals by increased *mel-46* levels ([*mel-46(+)*#1] rescue array, *Figure 2—figure supplement 1D*), suggesting a privileged relationship between SMN-1 and MEL-46 in cholinergic NMJ signaling. Increasing *mel-46* did rescue *smn-1(ok355)* synaptic protein localization defects. Using the [*mel-46(+)*#1] rescue array, we rescued *smn-1(ok355)* defective SNB-1 puncta width and intensity to normal levels, but did not ameliorate linear density defects (*Figure 2F–J*). Notably, increased *mel-46* in an *smn-1(+)* control background did not increase SNB-1 levels, suggesting that *mel-46*-induced up-regulation of SNB-1 is specifically beneficial in a *smn-1(ok355)* background. Increasing *mel-46* with a second broadly-expressed *mel-46* rescue array line, [*mel-46(+)*#2], also restored APT-4 puncta linear density to normal levels (*Figure 2—figure supplement 1E*), without rescuing other metrics. These results are consistent with the aldicarb/NMJ functional rescue studies presented here and suggest increasing *mel-46* improves neuromuscular signaling in *smn-1(ok355)* animals by partially restoring levels and localization of synaptic proteins. Furthermore, these results suggest that *mel-46* may act with or downstream of *smn-1* in a pathway essential for NMJ function.

Alternatively, we considered the possibility that increasing *mel-46* stabilizes maternally-loaded SMN-1 protein and mRNA, a scenario which could also explain *mel-46* rescue of *smn-1(ok355)* NMJ function. Since mammalian Gemin3 directly binds SMN (*Charroux et al., 1999*), increasing MEL-46 (Gemin3) might decrease the rate of SMN-1 loss. To test this possibility, we used CRISPR/Cas9-targeted genome editing to insert GFP coding sequences at the N-terminus of *smn-1* on chromosome I, resulting in fluorescent SMN-1 protein (*Figure 2—figure supplement 2A*) (*Dickinson et al., 2015*). The GFP-tagged protein was functional; animals were viable and fertile. We found that increasing *mel-46* using the [*mel-46(+)*#2] rescue array did not increase GFP::SMN-1 levels, but unexpectedly caused a modest overall decrease (*Figure 2—figure supplement 2B and C*). It therefore seems unlikely that *smn-1(ok355)* rescue by increased *mel-46* is due to stabilization of maternally-loaded SMN-1. Using the same CRISPR-based method, we were unable to tag endogenous *smn-1* on balancer chromosomes necessary for maintaining the *smn-1(ok355)* line. This approach would have allowed us to

examine the effects of MEL-46 overexpression on the stability of maternally-loaded SMN-1 in *smn-1 (ok355)* animals. Therefore, although we cannot rule out stability of maternally-loaded SMN-1 as a contributing factor, the large effect that decreased SMN-1 has on MEL-46 localization favors a mechanism in which MEL-46 overexpression rescues *smn-1(ok355)* defects, at least in part, by restoring MEL-46 functional deficits in this background.

## *C.elegans* miR-2 is required for NMJ function

The pathways in MNs downstream of SMN and Gemin3 that are linked to SMA are unknown. Here, we consider a role for these two proteins in miRNA regulation. As mammalian Gemin3 co-localizes and co-purifies with RISC pathway components (*Höck et al., 2007*; *Hutvágner and Zamore, 2002*; *Meister et al., 2005*; *Mourelatos et al., 2002*; *Murashov et al., 2007*), we considered a role for miRNA regulation in NMJ function. miRNA miR-2 is enriched in neurons, expressed at all developmental stages, and predicted to regulate expression of many proteins involved in neuronal development and function (*Marco et al., 2012*; *Martinez et al., 2008*). We hypothesized that miR-2 is necessary for proper NMJ function and that it might be perturbed by loss of either SMN-1 or MEL-46. To test this possibility, we first examined the aldicarb response of *mir-2(lf)* animals. Two different deletion alleles, *gk259* and *n4108*, caused resistance to aldicarb paralysis compared to wild type animals (*Figure 3A*; *Figure 3—figure supplement 1A*). This defect was partially rescued by expressing miR-2 under the control of a the *unc-17* (ACh) cholinergic neuron-specific promoter (referred to as ([ACh::*mir-2(+)*])) (*Figure 3A*). Loss of miR-2 also resulted in a mild pharyngeal pumping defect (*Figure 3—figure supplement 1B*). Taken together, we conclude that miR-2 is required in cholinergic neurons for proper NMJ signaling.

As a first step towards evaluating how miR-2 loss impacts cholinergic MN presynaptic function, we examined the effects of miR-2 loss on localization of presynaptic proteins in DA MNs. Four fluorescently-tagged presynaptic proteins were examined: SNB-1 (synaptobrevin), SYD-2 (α-liprin), ITSN-1 (DAP160/Intersectin), and APT-4 (AP2 α-adaptin). Analysis of tagged SNB-1 in *mir-2(gk259)* animals revealed increased SNB-1 puncta linear density, but no change in puncta width or intensity compared to wild type animals (*Figure 3B–D*). Additionally, *mir-2(gk259)* animals were indistinguishable from wild type control animals with respect to SYD-2 synaptic localization for all metrics analyzed (*Figure 3—figure supplement 1C*), suggesting that pre-synaptic active zones are unchanged in number and size; thus, synaptic changes are likely not the result of altered active zone number or size (*Zhen and Jin, 1999*). ITSN-1 puncta width and intensity, but not linear density, were decreased in *mir-2(gk259)* and *mir-2(n4108)*, animals compared to wild type controls (*Figure 3—figure supplement 1D–F*). Finally, both *mir-2(gk259)* and *mir-2(n4108)* had decreased APT-4 puncta width, intensity, and linear density (*Figure 3—figure supplement 1G–I*). ITSN-1, similar to APT-4, is involved in vesicle recycling at the NMJ (*Wang et al., 2008*). Together with results from aldicarb resistance studies, our results suggest that loss of miR-2 results in synaptic dysfunction at the NMJ, consistent with decreased cholinergic synaptic release (*Ch'ng et al., 2008*; *Sieburth et al., 2005*). Additionally, we observed considerable overlap between synaptic protein localization defects resulting from miR-2 loss with those of *smn-1(ok355)* animals (*Dimitriadi et al., 2016*), a finding consistent with miR-2 and SMN-1 acting in partially redundant pathways at the NMJ.

## *C.elegans* miR-2 targets *gar-2* mRNA in cholinergic neurons

To address mechanistically how miR-2 loss impacts NMJ function, we searched for mRNA targets of miR-2. Canonically, miRNA loss results in overexpression of direct mRNA targets (*Elbashir et al., 2001*). Since miR-2 loss leads to aldicarb resistance, loss of the target(s) is expected to cause hypersensitivity to aldicarb. Following a literature search for genes whose loss of function results in hypersensitivity, we selected the following genes with putative miR-2 3'UTR binding sites for study: *gar-2*, *dbl-1*, *sek-1*, and *vab-2* (*Jan et al., 2011*; *Lewis et al., 2005*; *Paraskevopoulou et al., 2013*; *Reczko et al., 2012*; *Vashlishan et al., 2008*). Loss of a *bona fide* target gene is predicted to suppress aldicarb resistance caused by miR-2 loss. Therefore, we crossed a deletion allele for each gene into the *mir-2(gk259)* background. Loss of any of these four genes suppressed *mir-2(gk259)* to some extent, but *gar-2(ok520)*, which contains a large deletion removing *gar-2* exons 6 and 7, resulted in the most complete suppression, thus suggesting that GAR-2 acts downstream of miR-2 (*Figure 4A*;

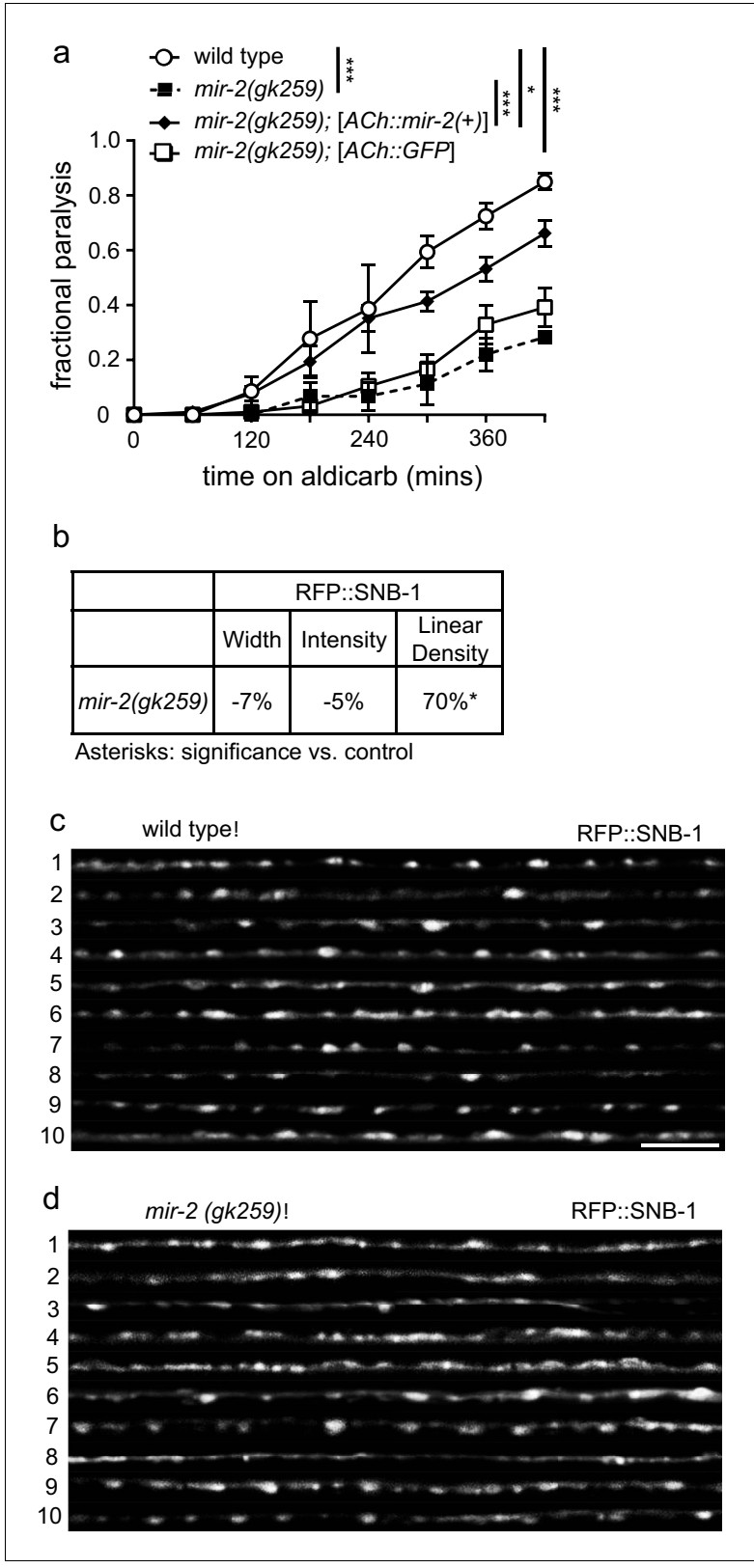

**Figure 3.** miR-2 is required in cholinergic neurons for proper NMJ function. (a) *mir-2(gk259)* animals were resistant to paralysis by aldicarb. Expression of miR-2 behind the *unc-17* (ACh) cholinergic promoter partially restored *mir-2 (gk259)* sensitivity to aldicarb compared to transgenesis controls expressing GFP behind the same promoter. Time course for paralysis on 1 mM aldicarb for wild type (N2), *mir-2(gk259)*, *mir-2(gk259);[ACh::mir-2(+)]* and *mir-2*

*Figure 3 continued on next page*

*Figure 3 continued*
(*gk259*);[*ACh::GFP*] young adult animals. Log-rank test. (**b**) *mir-2(gk259)* animals had increased RFP::SNB-1 (synaptobrevin) linear density. Percent change from wild type (N2) control for RFP::SNB-1 in the dorsal nerve cord of *mir-2(gk259)* animals for 'punctaanalyzer' parameters: puncta width (μm), intensity (AU), and linear density (number/μm). *t*-test, two-tailed. (**c–d**) Representative images of cholinergic DA MN RFP::SNB-1 in the dorsal cord of wild type and *mir-2(gk259)* animals. These images were taken as part of data collection. Scale bar, 5 μm.
The following source data and figure supplements are available for figure 3:

**Source data 1.** Raw Data for *Figure 3*.
**Figure supplement 1.** miR-2 is required for NMJ function.
**Figure Supplement 1—source data 1.** Raw Data for *Figure 3—figure supplement 1*.

*Figure 4—figure supplement 1A–C*). GAR-2 is a G protein-coupled acetylcholine receptor orthologous to the mammalian M2 muscarinic receptor (m2R) (*Lee et al., 2000*).

To determine if miR-2 regulates GAR-2 expression directly, we examined the consequences of perturbing the putative miR-2 binding site in the *gar-2* 3'UTR. Using CRISPR/Cas9-targeted genome editing, we scrambled the 18 base pair *gar-2* 3'UTR region corresponding to the endogenous miR-2 binding site (*Figure 4B*). Compared to control animals (*gar-2* UTRwt[C]) carrying the disrupted PAM site mutation (C>T), we found that animals with the scrambled miR-2 binding site (*gar-2* UTRscr[C]) were resistant to aldicarb, similar to miR-2 loss (*Figure 4C*). To evaluate whether disruption of the miR-2 binding site influenced *gar-2* transcript levels, we compared *gar-2* mRNA levels in *gar-2* UTRscr[C] animals and *gar-2* UTRwt[C] controls and found a 40% increase in *gar-2* transcript in *gar-2* UTRscr[C] (*Figure 4D*). Disruption of the 3'UTR site likely inhibits binding of other miR-2 family members, possibly contributing to the effect we observe (*Ibáñez-Ventoso et al., 2008*).

To test the effects of miR-2 loss on GAR-2 function in cholinergic neurons, we undertook in vivo GFP reporter analysis of GAR-2 expression. We generated a construct encoding GFP with a *gar-2* 3'UTR, whose expression is driven under the control of a cholinergic-specific promoter (referred to as *unc-17p*-ACh::GFP::*gar-2* 3'UTRwt). A second control version of the construct contained the same scrambled UTR sequence as used in *gar-2* UTRscr[C] animals (referred to as *unc-17p*-ACh::GFP::*gar-2* 3'UTRscr) (*Figure 4E*). Transgenic lines were created by multicopy insertion for each construct. Increased GFP levels were observed in *mir-2(gk259)* animals expressing the intact 3'UTR construct as compared to control animals (*Figure 4F*; *Figure 4—figure supplement 2A*). Whereas, loss of the *mir-2* gene did not affect GFP levels in animals expressing the scrambled 3'UTR construct compared to control (*Figure 4G*; *Figure 4—figure supplement 2B and C*). To quantify impact of miR-2 on expression of these GFP reporters in various genetic backgrounds, we compared relative changes in GFP levels of transgenic lines using a ratiometric strategy; we determined GFP expression for lines carrying *unc-17p*-ACh::GFP::*gar-2* 3'UTRwt and compared these values to lines carrying *unc-17p*-ACh::GFP::*gar-2* 3'UTRscr in the same genetic background (*Figure 4—figure supplement 2D*). By this method, we observed a ~13% increase in relative GFP expression associated with loss of miR-2 (*Figure 4H*), indicating that miR-2 directly suppresses translation of *gar-2* mRNA by binding the *gar-2* 3'UTR at this 3'UTR site. We also assessed the effect of miR-2 loss on *gar-2* transcript levels and found no significant difference in *gar-2* mRNA levels between wild type animals and *mir-2(gk259)* loss of function animals (*Figure 4I*). These results, combined with our reporter data, suggest that miR-2 binds and inhibits *gar-2* mRNA translation, but does not reduce transcript levels. Previous studies have reported that miRNAs can influence protein synthesis of targets without destabilizing mRNA levels (*Cloonan, 2015*; *Selbach et al., 2008*).

## miR-2 suppression of *gar-2* is disrupted by smn-1 loss

Next, we determined if *smn-1* loss altered miR-2 regulation of the *gar-2* 3'UTR. Both GFP reporter transgenes were crossed into the *smn-1(ok355)* background. Using the same ratiometric strategy from *Figure 4H*, we observed a small increase in relative GFP reporter expression (~5%) in *smn-1 (ok355)* animals compared to *smn-1(+)* controls (*Figure 5A*; *Figure 5—figure supplement 1A and B*).

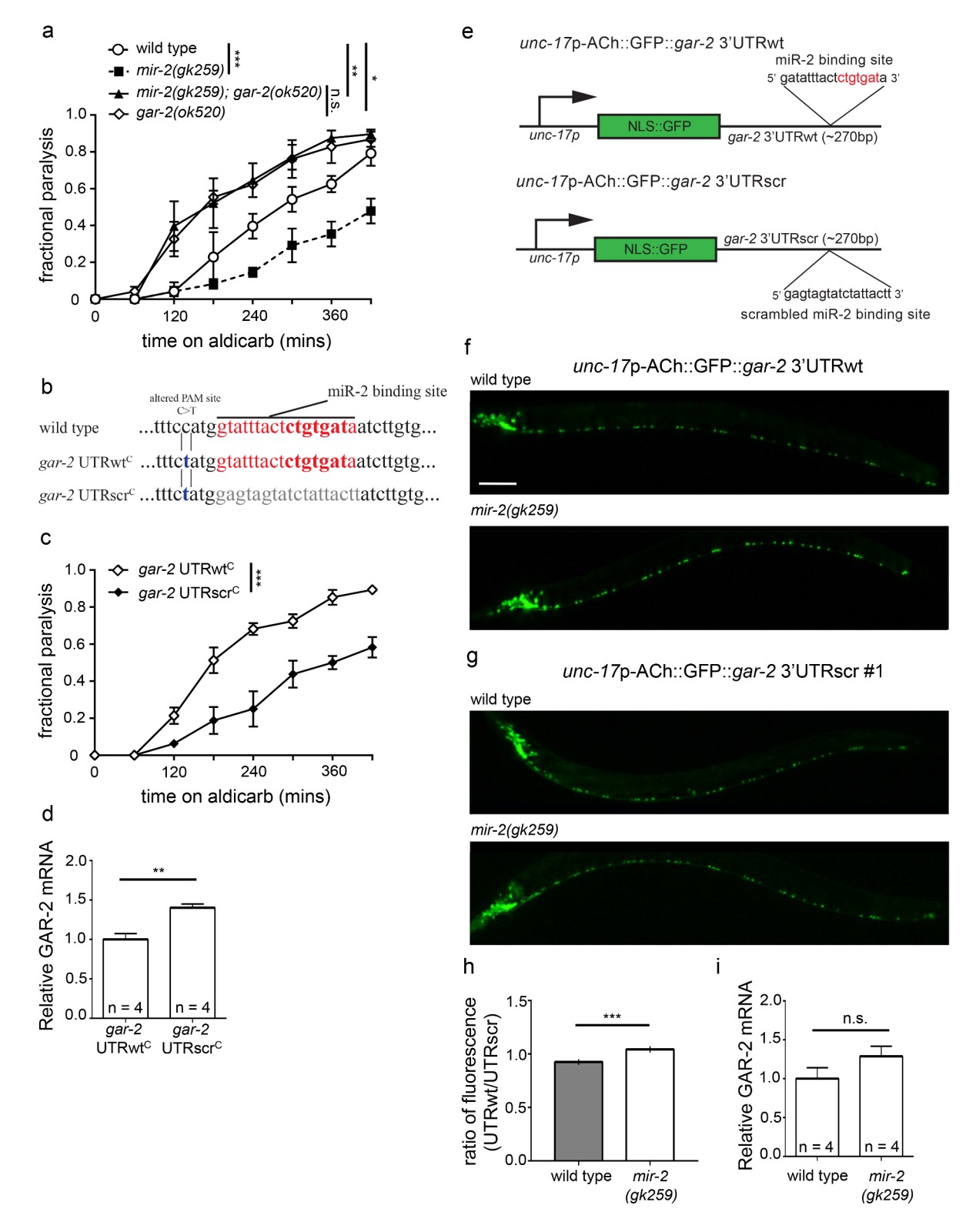

**Figure 4.** miR-2 binds the gar-2 3'UTR and represses GAR-2 translation. (**a**) Loss of m2R ortholog, GAR-2, suppressed aldicarb response defects of animals lacking *mir-2(gk259)*. *gar-2(ok520)* animals were hypersensitive to paralysis by aldicarb. Time course for paralysis on 1 mM aldicarb for wild type (N2), *mir-2(gk259)*, *gar-2(ok520)* and *mir-2(gk259);gar-2(ok520)* young adult animals. Log-rank test. (**b**) Schematic representation of changes made to the endogenous *gar-2* 3'UTR using CRISPR. For the wild type control (*gar-2* UTRwt[C]), the miR-2 binding site remained intact, however, a C>T PAM site
*Figure 4 continued on next page*

*Figure 4 continued*

change was made. For the experimental condition (*gar-2* UTRscr[C]), the miR-2 binding site was scrambled in addition to the C>T PAM site alteration. (**c**) *gar-2(rt318)*, referred to as *gar-2* UTRscr[C], animals were resistant to the acetylcholinesterase inhibitor, aldicarb, compared to *gar-2(rt317)*, referred to as *gar-2* UTRwt[C]. Time course for paralysis on 1 mM aldicarb for young adult animals. Log-rank test. (**d**) Scrambling the predicted endogenous *gar-2* 3'UTR miR-2 binding site increased *gar-2* messenger RNA levels. Quantification of *gar-2* mRNA levels in young adult *gar-2* UTRwt[C] and *gar-2* UTRscr[C] animals. *t*-test, two-tailed (n = 4 for *gar-2* UTRwt[C] and *gar-2* UTRscr[C]). (**e**) Reporter constructs used to assess miR-2 regulation of *gar-2* 3'UTR in cholinergic neurons: *rtIs56* (*unc-17*p-ACh::GFP::*gar-2* 3'UTRwt) and *rtIs57* or *rtIs58* (*unc-17*p-ACh::GFP::*gar-2* 3'UTRscr). *unc-17*p-ACh::GFP::*gar-2* 3'UTRwt construct contains the *unc-17* promoter expressing NLS::GFP upstream of the *gar-2* 3'UTR, which has a predicted miR-2 binding site. Red text indicates intact seed region. *unc-17*p-ACh::GFP::*gar-2* 3'UTRscr is the same construct with the predicted miR-2 binding site scrambled identically to the sequence in *gar-2* UTRscr[C] animals. (**f**) Representative images of *unc-17*p-ACh::GFP::*gar-2* 3'UTRwt expression in cholinergic neurons of wild type (N2) and *mir-2(gk259)* larval stage L4 animals. Scale bar, 50 µm. (**g**) Representative images of *unc-17*p-ACh::GFP::*gar-2* 3'UTRscr (*rtIs57*) expression in cholinergic neurons of wild type (N2) and *mir-2(gk259)* larval stage L4 animals. (**h**) Ratio representation of mean GFP fluorescence for wild type and *mir-2(gk259)* animals. *t*-test, two-tailed. Ratio was calculated by dividing the mean GFP fluorescence of *unc-17*p-ACh::GFP::*gar-2* 3'UTRwt for each genotype by the corresponding mean GFP fluorescence of *unc-17*p-ACh::GFP::*gar-2* 3'UTRscr for that genotype. UTRwt represents mean fluorescence for each genotype expressing the *unc-17*p-ACh::GFP::*gar-2* 3'UTRwt reporter, while UTRscr represents mean fluorescence for each genotype expressing the *unc-17*p-ACh::GFP::*gar-2* 3'UTRscr control reporter. Error bars represent the cumulative SEM for each genotype across transgenes. (see ***Figure 4— figure supplement 2D***). (**i**) *gar-2* transcript levels did not increase in a *mir-2* loss of function background. Quantification of *gar-2* mRNA levels in young adult *mir-2(gk259)* animals compared to wild type (N2) controls. *t*-test, two-tailed (n = 4 for *mir-2(gk259)* and N2).

The following source data and figure supplements are available for figure 4:

**Source data 1.** Raw Data for ***Figure 4***.

**Figure supplement 1.** Loss of predicted miR-2 mRNA targets suppresses *mir-2(lf)* aldicarb resistance.

**Figure Supplement 1—source data 1.** Raw Data for ***Figure 4—figure supplement 1***.

**Figure supplement 2.** miR-2 inhibits translation by binding the *gar-2* 3'UTR.

**Figure Supplement 2—source data 1.** Raw Data for ***Figure 4—figure supplement 2***.

This finding suggested that *smn-1(ok355)* animals may have decreased miR-2 function and, as a consequence, an increase in GAR-2 translation. We showed above that increased *mel-46* levels ameliorates *smn-1(ok355)* NMJ defects (***Figure 2***; ***Figure 2—figure supplement 1***), therefore we assessed the impact of increased *mel-46* on miR-2 activity using the [*mel-46(+)#2*] rescue array. In control *smn-1(+)* animals, increasing *mel-46* did not alter relative expression of the miR-2 GFP reporter. However, in *smn-1(ok355)* animals, increasing *mel-46* caused decreased relative reporter expression by ~15%, compared to *smn-1(ok355)* animals lacking the *mel-46* rescue array (***Figure 5A***; ***Figure 5—figure supplement 1A and B***). These data suggest that MEL-46 overexpression decreases GAR-2 levels in *smn-1 (ok355)* by increasing miR-2 activity.

Increased GAR-2 translation in animals lacking SMN-1 might be due to decreased mature miR-2 levels. To test this possibility, we used quantitative RT-PCR studies. After neuron-specific RNAi knock-down of either SMN-1 or MEL-46, we found decreases in mature miR-2 levels (***Figure 5B***; ***Figure 5—figure supplement 1C and D***), but no change in *gar-2* transcript levels (***Figure 5C***). This result is consistent with our finding that *gar-2* transcript levels were unchanged in miR-2 complete loss conditions (***Figure 4I***), despite alterations in GFP reporter expression (***Figure 4H***). These results suggest neuronal miR-2 levels are decreased when MEL-46 or SMN-1 levels decrease.

Collectively, our data above are consistent with a model where diminished SMN-1 leads to decreased miR-2 levels and activity, resulting in increased GAR-2 expression. Since M2 receptors inhibit synaptic release at cholinergic NMJs across species (***Dittman and Kaplan, 2008***; ***Parnas et al., 2005***; ***Slutsky et al., 2003***), overexpression of these receptors in *smn-1(ok355)* MNs might contribute to the NMJ defects previously observed in these animals (***Dimitriadi et al., 2016***; ***Sleigh et al., 2011***). Furthermore, increased MEL-46/Gemin3 might have an ameliorative effect in animals lacking SMN-1 by stimulating miR-2 activity, thus decreasing GAR-2 levels and disinhibiting cholinergic release.

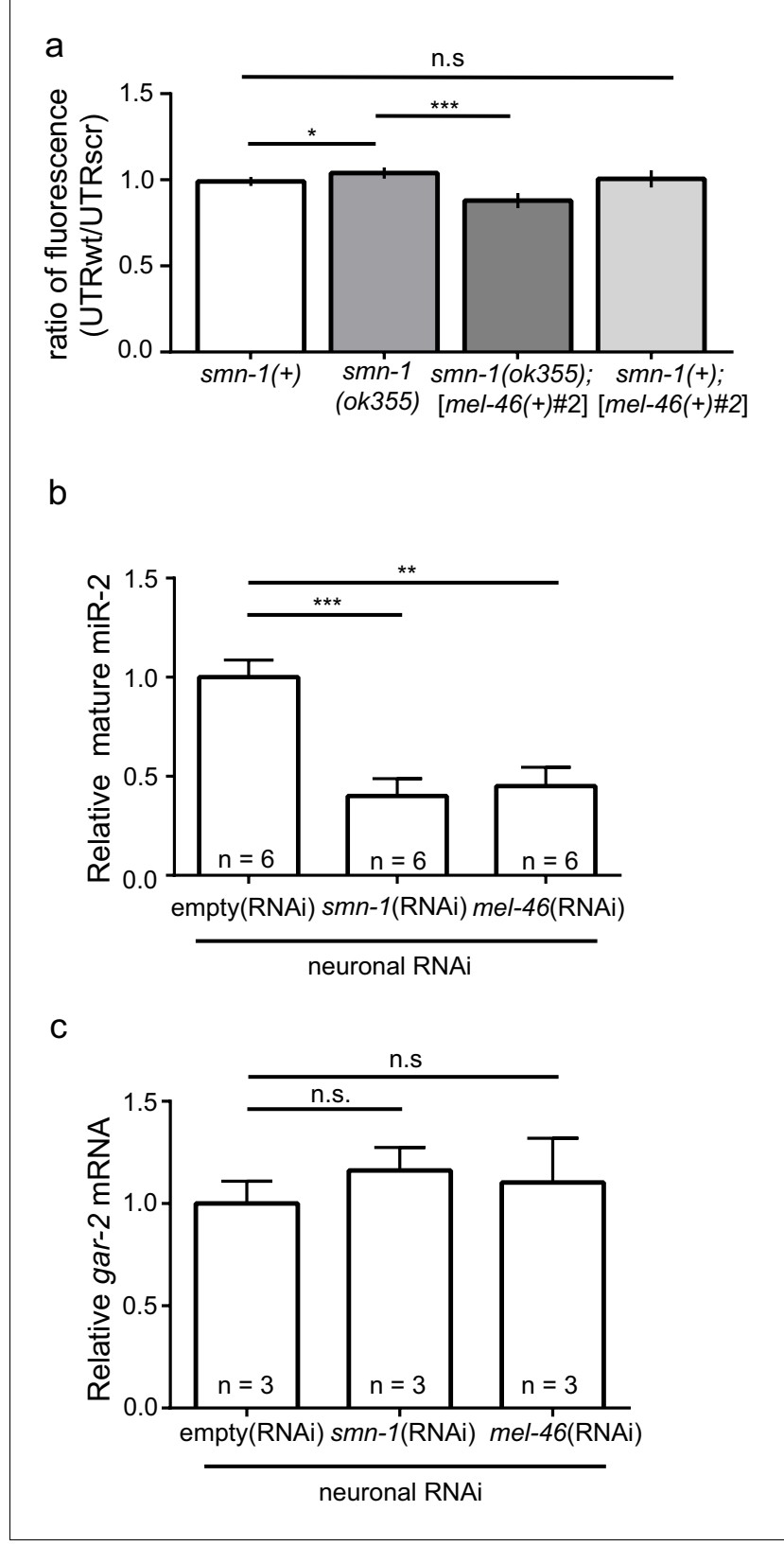

**Figure 5.** *smn-1* loss of function abrogated miR-2 repression of GAR-2 expression. (a) Loss of *smn-1* caused a relative increase in *unc-17*p-ACh::GFP::*gar-2* 3'UTRwt expression. Expressing *mel-46* using the broadly expressed [*mel-46(+)#2*] array decreased relative *unc-17*p-ACh::GFP::*gar-2* 3'UTRwt expression in *smn-1(ok355)* animals. Ratio

*Figure 5 continued on next page*

*Figure 5 continued*
representation of mean GFP fluorescence for *smn-1(+)*, *smn-1(ok355)*, *smn-1(ok355);[mel-46(+)#2]*, and *smn-1(+);*
*[mel-46(+)#2]* animals. Mann-Whitney *U*-test, two-tailed. Ratio calculation was completed in the same manner as
*Figure 4H* (see also *Figure 4—figure supplement 2D*) (b) miR-2 levels were decreased in neurons when either
SMN-1 or MEL-46 were decreased. Quantification of mature miR-2 for empty(*RNAi*), *smn-1(RNAi)*, and *mel-46*
*(RNAi)* young adult animals relative to housekeeping miRNA miR-60. *t*-test, two-tailed (n = 6 for each condition).
(c) *gar-2* transcript levels did not change when SMN-1 or MEL-46 levels decreased in neurons. Quantification of
*gar-2* mRNA for empty(*RNAi*), *smn-1(RNAi)*, and *mel-46(RNAi)* young adult animals. *t*-test, two-tailed (n = 3 for
each condition). (c–d) Animals sensitive to RNAi in only neurons (TU3401) were fed bacteria expressing double-
stranded RNA (dsRNA) against *mel-46* or *smn-1*. Control animals were fed bacteria expressing an empty vector
control: empty(*RNAi*).
The following source data and figure supplements are available for figure 5:

**Source data 1.** Raw data for *Figure 5*.
**Figure supplement 1.** Increasing MEL-46(Gemin3) ameliorates *smn-1(lf)* defective miR-2 activity.
**Figure Supplement 1—source data 1.** Raw Data for *Figure 5—figure supplement 1*.

## *gar-2* loss ameliorates *smn-1(lf)* neuromuscular defects

If increased GAR-2 levels exacerbate NMJ dysfunction in animals lacking SMN-1, then decreasing
GAR-2 function should ameliorate the NMJ defects caused by *smn-1* loss of function. Loss of GAR-2
did not improve pharyngeal pumping rates in *smn-1(ok355)* animals (*Figure 6—figure supplement
1A*), similar to our results in animals with increased *mel-46* levels (*Figure 2—figure supplement 1D*).
However, similar to increasing *mel-46*, *gar-2(ok520)* restored normal response to aldicarb in both
*smn-1(ok355)* and *smn-1(rt248)* animals (*Figure 6A*; *Figure 6—figure supplement 1B*), consistent
with improved NMJ function. The *rt248 smn-1* allele causes a frameshift and loss of SMN-1 function
similar to *ok355* (*Dimitriadi et al., 2016*). Additionally, *gar-2(ok520)* restored normal response to
aldicarb in *mel-46(tm1739)* animals (*Figure 6B*). Our results indicate that decreasing GAR-2 likely
improves presynaptic function in animals with decreased SMN-1 or MEL-46. Consistent with this con-
clusion, *gar-2(ok520)* also rescued numerous presynaptic protein localization defects caused by
SMN-1 loss; SNB-1 puncta width, intensity and linear density defects were rescued in both *smn-1
(ok355)* and *smn-1(rt248)* backgrounds (*Figure 6C–6G*; *Figure 6—figure supplement 1D–H*). GAR-2
loss in *smn-1(+)* control animals resulted in increased SNB-1 puncta width and intensity, but did not
alter SNB-1 puncta linear density.

Research in vertebrates has demonstrated m2R internalization normally occurs in response to
chronic m2R stimulation, either by pharmacological agonist application or acetylcholinesterase inhi-
bition (*Clancy et al., 2007*). Since endocytosis is defective in animals lacking SMN-1, perturbed
endocytosis, in combination with miRNA misregulation, could contribute to GAR-2 accumulation at
the membrane leading to decreased SNB-1 in *smn-1(ok355)* motor neurons (*Dimitriadi et al., 2016*).
Loss of GAR-2 did not rescue *smn-1(ok355)* APT-4 puncta defects (*Figure 6—figure supplement
1C*), but decreased APT-4 puncta width and intensity in *smn-1(+)* control animals. As loss of GAR-2
did not restore APT-4 synaptic defects, we conclude that there are additional pathways affected by
*smn-1* loss, beyond GAR-2 misregulation. Nevertheless, these results suggest that *C. elegans* GAR-2
levels are increased by *smn-1* loss, which might contribute to NMJ defects in *smn-1(lf)* animals.

## The GAR-2 mammalian ortholog, m2R, is increased in SMA mouse model motor neurons

The closest human ortholog of GAR-2 is the M2 muscarinic receptor (m2R), encoded by the *CHRM2*
gene. GAR-2 and m2R are functionally conserved, as activation of these presynaptic receptors by
acetylcholine in different species results in hyperpolarization and decreased NMJ acetylcholine
release across species (*Dittman and Kaplan, 2008*; *Dudel, 2007*; *Parnas et al., 2005*;
*Slutsky et al., 2003*). Previous research suggests decreased SMN function across species might
impact miRNA activity across species, which could increase m2R levels consistent with our work in *C.*

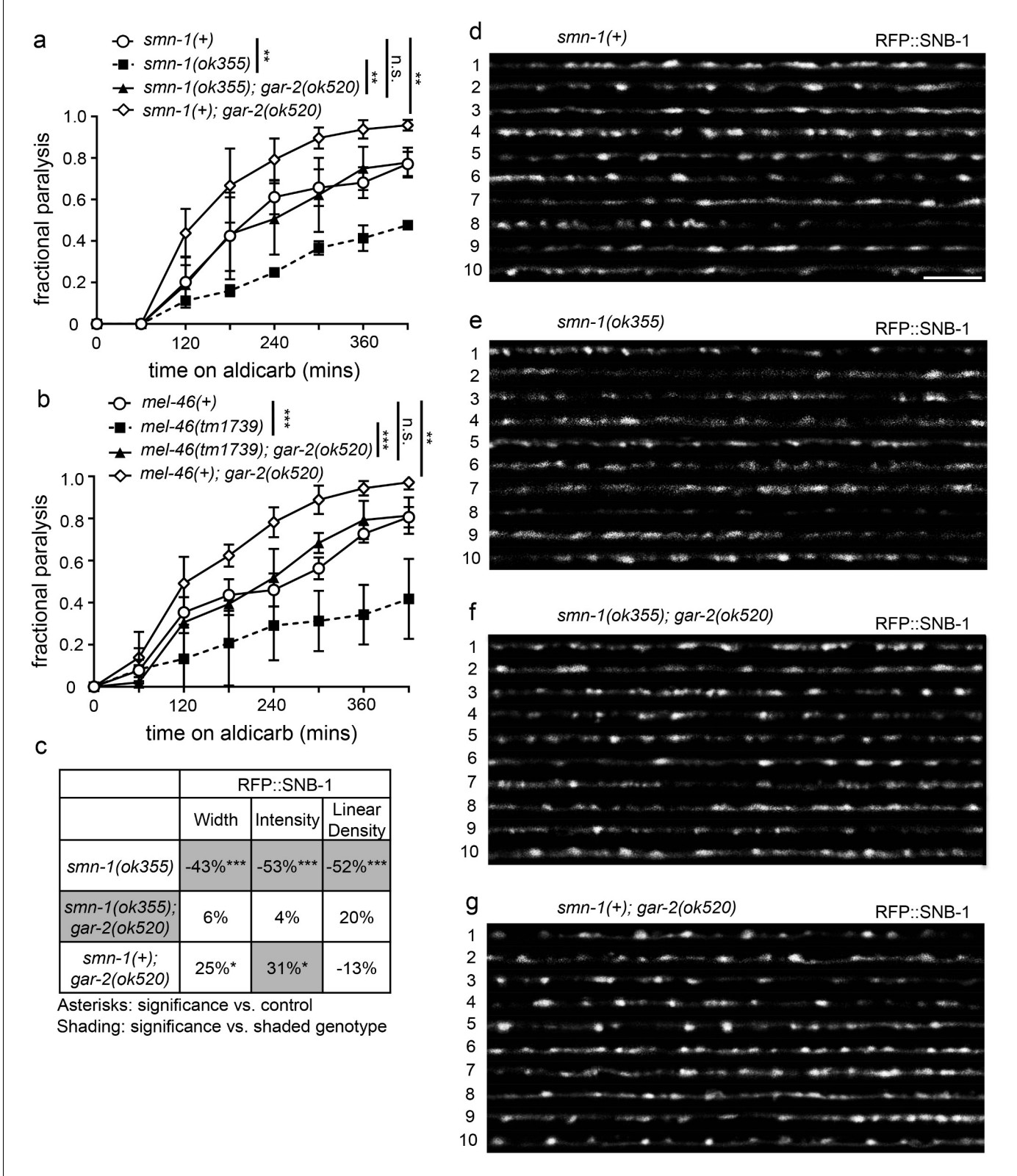

**Figure 6.** Loss of *gar-2* ameliorated *smn-1(lf)* NMJ defects. (**a**) Loss of *gar-2* rescued *smn-1(ok355)* aldicarb response defect. Time course for paralysis on 1 mM aldicarb for *smn-1(+)*, *smn-1(ok355)*, *smn-1(ok355);gar-2(ok520)*, and *smn-1(+);gar-2(ok520)* early larval stage L4 animals. Log-rank test. (**b**) Loss of *gar-2* rescued *mel-46(tm1739)* aldicarb response defect. Time course for paralysis on 1 mM aldicarb for *mel-46(+)*, *mel-46(tm1739)*, *mel-46(tm1739); gar-2(ok520)*, and *smn-1(+);gar-2(ok520)* early larval stage L4 animals. Log-rank test. For these experiments, *mel-46(tm1739)* was maintained over the

*Figure 6 continued on next page*

*Figure 6 continued*

*nT1* balancer and therefore, control *mel-46(+)* animals were obtained as +/+ animals from +/*nT1* heterozygous mothers. (c) Loss of *gar-2* rescued *smn-1 (ok355)* RFP::SNB-1 synaptic localization defects. *gar-2* loss in the *smn-1(+)* background resulted in increased RFP::SNB-1 puncta width and intensity. Percent change from *smn-1(+)* control for RFP::SNB-1 in the dorsal nerve cord of *smn-1(ok355)*, *smn-1(ok355);gar-2(ok520)*, and *smn-1(+);gar-2(ok520)* animals for 'punctaanalyzer' parameters: puncta width (μm), intensity (AU), and linear density (number/μm). Asterisks denote significance compared to *smn-1(+)* control; shading indicates significant difference from *smn-1(ok355);gar-2(ok520)*. Mann-Whitney *U*-test, two-tailed. Loss of *gar-2* rescued RFP:: SNB-1 puncta width defects (*smn-1(+)* control animals *versus smn-1(ok355);gar-2(ok520)* p=0.76; *smn-1(ok355)* versus *smn-1(ok355);gar-2(ok520)* p=0.0001), restored SNB-1 puncta intensity (*smn-1(+)* control animals *versus smn-1(ok355);gar-2(ok520)* p=1.00; *smn-1(ok355)* versus *smn-1(ok355);gar-2 (ok520)* p=0.0001) and rescued SNB-1 puncta linear density defects (*smn-1(+)* control animals *versus smn-1(ok355);gar-2(ok520)* p=0.08; *smn-1(ok355)* versus *smn-1(ok355);gar-2(ok520)* p=0.02). (d–g) Representative images of cholinergic DA MN RFP::SNB-1 in the dorsal nerve cord of *smn-1(+)*, *smn-1 (ok355)*, *smn-1(ok355);gar-2(ok520)*, and *smn-1(+);gar-2(ok520)*. These images were taken as part of data collection. Scale bar, 5 μm. Figures D and G are also shown in *Figure 6—figure supplement 1* since this control data was collected alongside both *ok355* and *rt248* RFP::SNB-1 data.

The following source data and figure supplements are available for figure 6:

**Source data 1.** Raw Data for *Figure 6*.
**Figure supplement 1.** Decreasing GAR-2(m2R) levels rescues NMJ defects in *smn-1(lf)* and *mel-46(lf)* animals.
**Figure Supplement 1—source data 1.** Raw Data for *Figure 6—figure supplement 1*.

*elegans.* The *CHRM2* mRNA is a predicted target of miR-128 in mice and humans (*Figure 7A*) (*Jan et al., 2011*; *Lewis et al., 2005*; *Paraskevopoulou et al., 2013*). Based on results from *C. elegans*, we predicted that vertebrate SMN loss might disrupt miR-128 activity, also leading to increased m2R. m2R protein levels were examined in MNs isolated from E13.5 SMA mice (*Smn*[-/-]; *SMN2*[tg/0]). A ~50% increase in m2R levels was observed, compared to wild type control MNs (*Figure 7B and C*). miR-128 levels in SMA mouse MNs were decreased compared to wild type (*Figure 7D*). Combined, these results suggest that diminished SMN protein causes decreased levels of mature miR-128, thus disinhibiting m2R expression in MNs across species.

## Inhibition of m2R by methoctramine rescued axon outgrowth defects in SMA mouse model MNs

Decreased SMN levels results in axon outgrowth defects in MNs derived from a SMA mouse model (*Rossoll et al., 2003*) and increased m2R might contribute to this functional defect. To test this, we examined the impact of m2R pharmacological inhibition on axon length for DIV5 MNs from E13.5 wild type (FVB) and SMA mice (*Smn*[-/-];*SMN2*[tg/0]) (*Figure 7E*). Wild type and SMA MNs were cultured in the presence of 50 nm or 500 nm methoctramine, an m2R antagonist. In wild type MNs, methoctramine decreased mean longest axon length. Conversely, methoctramine treatment in SMA MNs increased both mean longest axon length and total axon length (*Figure 7E*; *Figure 7—figure supplement 1A*). We conclude that m2R inhibition rescues MN axon outgrowth defects in a SMA mouse model, consistent with a deleterious impact of increased m2R activity in SMA model MNs.

## Discussion

Our understanding of the mechanisms by which SMN loss influences MN function and survival remains incomplete. Many pathways are perturbed by SMN loss, including spliceosome assembly, pre-mRNA splicing, mRNA transport, endocytosis, and axon growth (*Dimitriadi et al., 2016*; *Fallini et al., 2011*; *Hosseinibarkooie et al., 2016*; *McWhorter et al., 2003*; *Meister et al., 2001*; *Pellizzoni et al., 2002*). As the human *CHRM2* locus lacks introns (*Gosso et al., 2007*), spliceosomal defects caused by SMN1 loss of function are unlikely to directly affect mature *CHRM2* mRNA levels, although indirect effects are possible. Based on our results, the simplest model explaining why SMN loss alters m2R levels is one in which decreased SMN levels impact Gemin3 function, thus perturbing miRNA pathways, including miR-128 suppression of m2R translation. As m2R receptors inhibit cholinergic MN synaptic release (*Slutsky et al., 2003*), m2R up-regulation may contribute to MN defects in early stages of SMA pathogenesis, in parallel to perturbation of other pathways affected by SMN loss (*Figure 7F*).

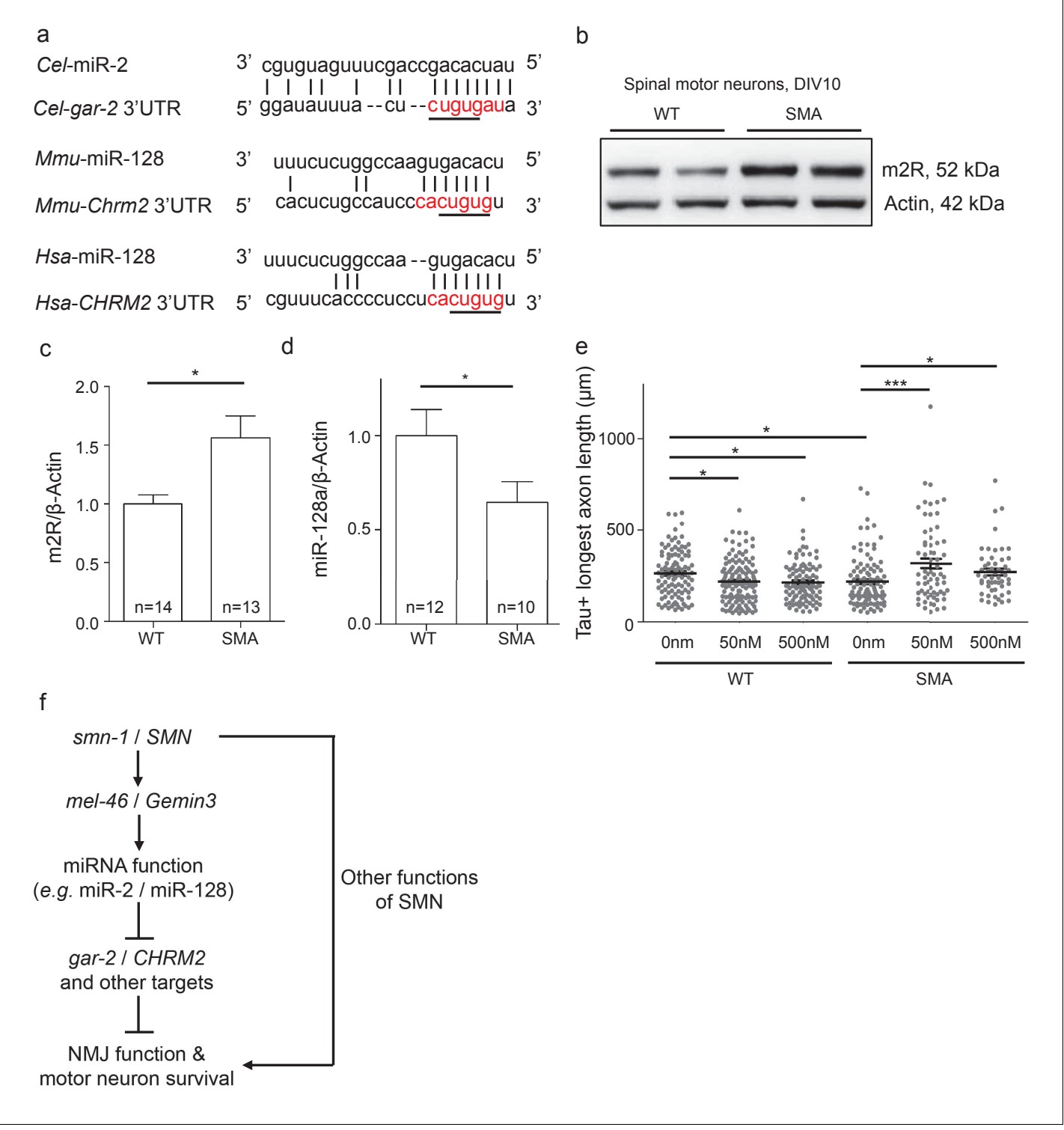

**Figure 7.** Increased m2R muscarinic receptor levels in SMA mouse model MNs contribute to axon outgrowth defects. (**a**) Alignment of predicted miR-2 or miR-128 binding sites for *C. elegans,* mouse and human *gar-2* or *CHRM2* 3'UTRs. *CHRM2* encodes the mR2 muscarinic receptor (***Paraskevopoulou et al., 2013***; ***Reczko et al., 2012***). Predicted nucleotide pairing shown by vertical lines. Red text indicates predicted miRNA seed region. A black line indicates potential seed region conservation. (**b**) Representative image for two E13.5 wild type and two *Smn*[-/-];*SMN2*[tg/0] DIV10 spinal MN immunoblots probed for m2R and control *β*-Actin. (**c**) Quantification of immunoblot, *t*-test, two-tailed, p=0.017 (n = 14 for WT and n = 13 for SMA). (**d**) Quantification of miR-128 levels in DIV10 spinal MNs from E13.5 wild type and *Smn*[-/-];*SMN2*[tg/0] animals. *t*-test, two-tailed (n = 24 from 12 mice for WT; n = 20 from 10 mice for SMA). (**e**) Longest axon length for E13.5 wild type and *Smn*[-/-];*SMN2*[tg/0] DIV5 spinal MNs treated with 0 nm, 50 nm, and

*Figure 7 continued on next page*

Figure 7 continued

500 nm methoctramine. *t*-test, two-tailed. (n = 103 for WT 0 nM, n = 131 for WT 50 nM, n = 98 for WT 500 nM, n = 102 for SMA 0 nM, n = 67 for SMA 50 nM and n = 53 for SMA 500 nM) (f) Proposed model: SMN acts via MEL-46 to influence microRNAs that play important roles in NMJ function and MN survival. SMN loss affects other pathways as well. *C. elegans* genes shown on left side; human genes on right side.

The following figure supplement is available for figure 7:

**Figure supplement 1.** m2R inhibition by methoctramine increases axon length in SMA mouse model MNs.

## SMN-1 loss impacts miRNA activity in *C. elegans* potentially via MEL-46/Gemin3

The most direct molecular connection between SMN and the miRNA pathway is Gemin3. Results presented here indicate that decreases in SMN-1(SMN) or MEL-46(Gemin3) result in similar neuro-muscular defects as well as decreased miR-2 levels in neurons. This analysis further suggests that decreases in *C. elegans* SMN-1 result in increased GAR-2 (m2R) through miR-2 misregulation and that increasing MEL-46 (Gemin3) reduces GAR-2 protein levels in *smn-1(ok355)* animals *via* modulation of miR-2. Several lines of evidence support this conclusion, including miR-2 GFP reporter expression results, NMJ aldicarb sensitivity, synaptic protein localization changes, and qPCR measurements of mature miR-2 and *gar-2* mRNA levels (*Figures 1–5*).

There is a paucity of research addressing the functional importance of Gemin3, in either the Gemin or the RISC complexes. Biochemical and co-localization studies support a role for Gemin3 in spliceosome assembly, mRNA transport, and miRNA function (*Charroux et al., 1999*; *Dostie et al., 2003*; *Feng et al., 2005*; *Mourelatos et al., 2002*; *Zhang et al., 2006*). It is possible that decreased and/or mislocalized Gemin3 impairs RISC function, leading to the diminished miR-2 activity and levels reported in *Figure 5*. However, since Gemin3 likely functions in numerous pathways, it may influence miR-2 through other, more indirect pathways. In *Drosophila*, Gemin3 is necessary for motor function, but the molecular mechanisms underlying this defect are unclear (*Cauchi et al., 2008*). Results presented here in *C. elegans* further support a requirement for Gemin3 in motor function and additionally, show that Gemin3 is capable of modifying miRNA levels and activity.

## SMN depletion results in miRNA misregulation across species

miR-2 belongs to the invertebrate K box family (motif: CUGUGAUA) of miRNAs (*Ibáñez-Ventoso et al., 2008*). It was suggested previously that miR-2 does not have well-conserved mammalian orthologs (*Marco et al., 2012*), but another study suggested that human miR-128 is a member of this miRNA family (*Ibáñez-Ventoso et al., 2008*). These two bioinformatics studies differ in their definition of the miRNA binding site, also known as the seed region. Our alignment of miR-2/miR-128 miRNAs and *gar-2/CHRM2* mRNAs suggests that the CUGUG seed region may be a conserved motif for miRNA binding (*Figure 7A*). Both miR-2 and miR-128 are enriched in the nervous system and share conserved mRNA targets (*Marco et al., 2012*; *Martinez et al., 2008*). More studies will be needed to confirm that miR-2 and miR-128 are orthologs. Regardless, the results presented here, in conjunction with previous research, suggest that overall miRNA misregulation contributes to neuronal defects when SMN levels decrease (*Haramati et al., 2010*; *Kye et al., 2014*; *Valsecchi et al., 2015*; *Wang et al., 2014*; *Wertz et al., 2016*).

Altered function of miR-2/miR-128 or GAR-2/m2R is not sufficient to explain all of the dysfunction observed in models of SMN deficiency. In *C. elegans,* miR-2 loss does not cause overt defects and GAR-2 loss does not restore viability, fecundity, or normal development to animals lacking SMN-1 or MEL-46. This suggests miR-2 loss does not contribute to defects outside the NMJ caused by SMN-1 loss. And, at the NMJ, synaptic protein perturbations are more severe in animals with diminished SMN-1 or MEL-46, compared to those lacking either miR-2 or GAR-2, which is consistent with a broader range of defects caused by decreased SMN-1 levels. In mice, complete miR-128 knock-out results in decreased motor activity and premature death (*Tan et al., 2013*), but it is currently unknown how miR-128 loss might specifically impact cholinergic MN function.

## M2 receptor expression is increased in SMA model MNs

We found that m2R levels were increased 50% overall in SMN-deficient mouse MNs compared to wild type by Western blot analysis (*Figure 7B and C*). In *C. elegans*, relative GAR-2 translation was increased by only 5% globally when SMN-1 levels dropped, based on GFP reporter expression in cholinergic neurons (*Figure 5A*). These two assays are not directly comparable, since the *C. elegans* experiment only investigated the contribution of miR-2 perturbation in regards to increased GAR-2 translation. Certainly, additional pathways are affected by SMN loss, beyond miRNAs, contributing to the greatly increased m2R levels in SMN-deficient MNs. These pathways may include mRNA transport metabolism, endocytosis, as well as spliceosome and ribonucleoprotein assembly (*Dimitriadi et al., 2016*; *Fallini et al., 2011*; *Hosseinibarkooie et al., 2016*; *Pellizzoni et al., 2002*; *Yong et al., 2002*). Extensive analysis would be required to understand how defects in these pathways contribute to increased m2R in SMN-deficient MNs.

## Could increased M2 muscarinic receptor expression contribute to α-MN defects in SMA patients?

m2R is expressed in α-MNs, with little to no expression in smaller gamma MNs (*Welton et al., 1999*). This expression profile correlates with the pattern of neurodegeneration in an SMA mouse model: selective loss of α-MNs, while gamma MNs remain unaffected (*Powis and Gillingwater, 2016*). Within α-MNs, m2R is distributed along the membrane and concentrates at postsynaptic connections with C-boutons (*Deardorff et al., 2014*). α-MNs in SMNΔ7 mice have increased C-bouton sites (*Tarabal et al., 2014*), which could be an additional cause or consequence of increased m2R levels in α-MNs. Taken together, this evidence suggests that increased m2R is consistent with multiple features of SMA pathology. We also consider three additional previously defined m2R pathways as possible contributors to MN functional defects in SMN-deficient α-MNs: GIRK channels, $Ca^{2+}$ channels, and SK channels.

Classically, m2R receptor activation leads to GIRK-channel-dependent efflux of $K^+$ cations, resulting in neuronal hyperpolarization and decreased SV release (*Sun et al., 2013*). Therefore, increased m2R levels are consistent with the synaptic defects observed in SMA models across species (*Dimitriadi et al., 2016*; *Kong et al., 2009*). m2R activation of GIRK channels is conserved in *C. elegans* (*Lee et al., 2000*), suggesting GAR-2 loss may rescue NMJ defects in animals lacking SMN-1 by reducing GIRK channel activation. Interestingly, sustained GIRK activation has been previously linked to neurodegeneration (*Coulson et al., 2008*).

m2R inhibits N-type $Ca^{2+}$ ($Ca_v2.2$) and P/Q-type $Ca^{2+}$ ($Ca_v2.1$) channels resulting in decreased SV release at the NMJ (*Slutsky et al., 2003*; *Yan and Surmeier, 1996*). And, $Ca_v2.1$-deficient mice exhibit NMJ degeneration and decreased active zone proteins (*Fox et al., 2007*; *Nishimune et al., 2004*). Additionally, in a mouse model of SMA, distal axons and growth cones had reduced $Ca^{2+}$ transients resulting from defective $Ca_v2.2$ excitability and accumulation (*Jablonka et al., 2007*). Increased m2R is consistent with decreased $Ca^{2+}$ channel activity observed in SMN-deficient animals. Moreover, $Ca_v2.2$ channels activate SK channels in α-MNs, suggesting increased m2R may lead to decreased SK channel currents (*Goldberg and Wilson, 2005*). The drug riluzole, which ameliorated motor defects in *smn-1(ok355)* animals, may act *via* SK channels (*Dimitriadi et al., 2013*). Increased m2R levels may result in excessive inhibition of SK channels, contributing to defective synaptic transmission in SMA models across species; this connection may offer additional mechanistic insight into the ameliorative effects of riluzole.

Previous reports suggest that decreases in $Ca^{2+}$ transients hinder axon outgrowth (*Hutchins and Kalil, 2008*). SMN loss also decreases these currents (*Jablonka et al., 2007*), consistent with defective axon outgrowth in SMN-deficient cultured neurons. Here, we show that inhibition of m2R by methoctramine ameliorates axon outgrowth defects in SMA mouse model MNs. As we find m2Rs are overexpressed in SMA MNs, methoctramine rescue of axon outgrowth may be the result of restored $Ca^{2+}$ channel function. Taken together, these data suggest that increased m2R expression contributes to axon outgrowth defects in SMA MNs and that m2R inhibition promotes axon outgrowth in SMA MNs.

We connect SMN functionally and mechanistically to the miRNA pathway. As an exemplar of this connection in two species, we demonstrate that decreased SMN levels lead to downregulation of specific miRNAs and consequent increased expression of M2 muscarinic receptors. Increased m2R activity is deleterious and consistent with a subset of the NMJ defects seen in SMA models, across

species. We suggest future studies might address the possible benefits of m2R inhibition in SMA models, as a combinatorial approach with other therapies.

## Materials and methods

### *C. elegans* strains, constructs and transgenes

Strains listed in *Supplementary file 2A* were maintained under standard conditions at 25°C (*Brenner, 1974*); we provide complete genotypes with unique strain identifiers, consistent with the rigorous standards of the *C. elegans* community. Abbreviated names are sometimes used for arrays, integrated lines or alleles in *Figures 1–5*; additional information about abbreviations can be found in *Supplementary file 2C*. For experiments with *smn-1(ok355)* and *smn-1(rt248),* animals assayed were first generation progeny of hermaphrodites heterozygous for the *hT2* balancer. To maintain a common genetic background, control *smn-1(+)* animals were also derived from +/*hT2* parents. Similarly, for APT-4::GFP synaptic localization (*Figure 2—figure supplement 1E*) and aldicarb response studies (*Figure 6B*), *mel-46(tm1739)* animals were first generation progeny of parents heterozygous for the *nT1* balancer. Control *mel-46(+)* animals were derived from +/*nT1* animals. For all other assays involving *mel-46(tm1739)*, animals were first generation progeny of parents carrying the *ytEx211[mel-46(+)]* rescue array; animals tested did not carry the array unless specified. For these experiments, N2 animals served as wild type controls. We attempted to generate *smn-1(ok355);mel-46(tm1739)* double mutant animals, but generation of heterozygous double mutant animals was not possible, using either balancer chromosomes or the *ytEx211[mel-46(+)]* rescue array.

The pHA#756 (*unc-17p::mir-2::unc-54* 3'UTR) plasmid was generated by excising a 867 bp fragment from pHA#755 (*aex-3p::mir-2::unc-54* 3'UTR) using NheI and SpeI. This fragment, containing the genomic *mir-2* pre-miRNA sequence along with *unc-54* 3'UTR sequence, was subcloned into pPD95.77 (pPD95.77 was a gift from Andrew Fire; Addgene, Cambridge, Massachusetts plasmid 1495) between NheI and SpeI sites, resulting in removal of the GFP sequence. Additionally, a 4466 bp fragment corresponding to the *unc-17* promoter was inserted between pPD95.77 SphI and AscI sites. Information for all amplification primers can be found in *Supplementary file 2B*. pHA#757 (*unc-17p::GFP::unc-54* 3'UTR) was generated by inserting the *unc-17* promoter fragment between pPD95.77 SphI and AscI sites, without altering the GFP sequence. Plasmid pHA#758 (NLS::GFP::*gar-2* 3'UTRwt) contains a 269 bp fragment corresponding to the *gar-2* 3'UTR that was subcloned into pPD95.67 (pPD95.67 was a gift from Andrew Fire; Addgene plasmid 1490) as a EcoRI and SpeI product. pHA#759 (*unc-17p::NLS::GFP::gar-2* 3'UTRwt) was generated by excising a 1286 bp fragment containing the NLS::GFP sequence and *gar-2* 3'UTR from pHA#758 using MscI and SpeI and ligating this fragment into pHA#756, thus removing the genomic *mir-2* pre-miRNA and *unc-54* 3'UTR sequences. pHA#760 was generated by ligating the *gar-2* 3'UTR fragment into pBluescript KS + (Stratagene, La Jolla, California) using EcoRI and SpeI. To construct pHA#761, the last 85 bp of the *gar-2* 3'UTR were removed from pHA#760 using NcoI and SpeI. This fragment was replaced with an identical 85 bp sequence, but with 19 bp scrambled at the predicted miR-2 binding site sequence (*gar-2* 3'UTRscr). Primers were annealed to produce this 85 bp sequence (*Supplementary file 2C*). Plasmid pHA#762 (NLS::GFP::*gar-2* 3'UTRscr) was generated by subcloning the 269 bp *gar-2* 3'UTRscr fragment from pHA#761 into pPD95.67 with EcoRI and SpeI. pHA#763 (*unc-17p::NLS:: GFP::gar-2* 3'UTRscr) was produced by subcloning the 1286 bp fragment containing NLS::GFP and *gar-2* 3'UTRscr sequences from pHA#762 into pHA#756 using MscI and SpeI, while removing the genomic *mir-2* pre-miRNA and *unc-54* 3'UTR sequences. pHA#790 (*unc-122p::mel-46::unc-54* 3'UTR) was created by amplifying the MEL-46 coding region from the pRM8 plasmid (*Minasaki et al., 2009*) and inserting this fragment into the pHA#729 EcoRI site (*Dimitriadi et al., 2016*). Using SphI and MscI restriction enzymes, the *unc-122* promoter was then excised and replaced with the 4466 bp *unc-17* promoter fragment excised from pHA#763 with the same enzymes, thus generating pHA#791 (*unc-17::mel-46::unc-54* 3'UTR). To create pHA#792 (*unc-17p::mel-46*::GFP::*unc-54* 3'UTR), a 906 bp GFP sequence was amplified from pHA#763 and subcloned by Gibson assembly into pHA#791 just before the MEL-46 stop codon (TGA). The small guide RNA (sgRNA) plasmids targeting the *smn-1* gene (pHA#764 and pHA#765) and the sgRNA plasmid targeting the *gar-2* 3'UTR (pHA#793) for CRISPR/Cas9-mediated genome editing were produced by amplification of *PU6::klp-12* (*Friedland et al., 2013*). Plasmid pHA#766 contains a GFP insertion template and self-excising

cassette flanked by *smn-1* arms of homology that were subcloned by Gibson assembly into pDD282 following a protocol from (*Dickinson et al., 2015*).

Integrated arrays *rtIs64* and *rtIs65 [unc-17p::mel-46::GFP::unc-54 3'UTR]* were created by UV irradiation of *rtEx871*, which were generated by standard injection of pHA#792 at 50 ng/μl, alongside 5 ng/μl myo-3p::mCherry (pCFJ104 - myo-3p::mCherry::unc-54utr was a gift from Erik Jorgensen (*Frøkjaer-Jensen et al., 2008*), and 75 ng/μl pBluescript KS+. To generate *rtEx855*[pRM8(*mel-46*(+)); *myo-2p*::RFP], wild type animals were injected with 133 ng/μl PRM8 plasmid (*Minasaki et al., 2009*), 5 ng/μl myo-3p::mCherry; Addgene plasmid 19328) and 75 ng/μl pBluescript KS+. Animals injected with rtEx855[pRM8(*mel-46*(+); *myo-3p*-RFP)] were crossed into a *mel-46(tm1739)* background to assure rescue of viability before further experiments were undertaken. Notably, expression of either *rtEx855* or *ytEx211* in *smn-1(lf)* animals did not rescue lethality or adult survival, further emphasizing a privileged relationship between SMN-1 and MEL-46 in cholinergic NMJ signaling. Lines for *rtEx853 [unc-17p::mir-2; myo-2p::mCherry]* and *rtEx854[unc-17p::GFP; myo-2p::mCherry]* were produced by injecting *mir-2(gk259)* animals with pHA#756 or pHA#757, respectively, at 40 ng/μl alongside 2.5 ng/μl myo-2p::mCherry (pCFJ90 - myo-2p::mCherry::unc-54utr was a gift from Erik Jorgensen (*Frøkjaer-Jensen et al., 2008*); Addgene plasmid 19327) and 77.5 ng/μl pBluescript KS+. *rtIs56[unc-17p:: GFP::gar-2 3'UTRwt; myo-2p::mCherry]* was integrated by UV irradiation into the genome and is derived from extrachromosomal array *rtEx856*, containing pHA#759, which was injected into wild type animals at 20 ng/μl with 2.5 ng/μl myo-2p::mCherry and 77.5 ng/μl pBluescript KS+. Integrated arrays *rtIs57* and *rtIs58 [unc-17p::GFP::gar-2 3'UTRscr; myo-2p::mCherry]* are two separate lines generated by UV irradiation of extrachromosomal array *rtEx857*, containing pHA#763, which was injected into wild type animals at 20 ng/μl with 2.5 ng/μl myo-2p::mCherry and 77.5 ng/μl pBluescript KS+. *gar-2(rt317)* and *gar-2(rt318)* alleles were generated by injecting *pha-1(e2123)* animals with the pHA#793 sgRNA plasmid targeting the *gar-2* 3'UTR at 25 ng/μl with either 50 ng/μl of a mutant single-strand oligo DNA (ssODN) repair template (*rt318*) or a control ssODN repair template (*rt317*), alongside the injection cocktail as described in *Ward (2015)*. Progeny from this injection were screened as described (*Ward, 2015*). Information on ssODN template sequences can be found in *Supplementary file 2C*. To generate *smn-1(rt280)*, which contains a GFP N-terminal insertion, wild type animals were injected with both pHA#764 and pHA#765 sgRNA plasmids targeting *smn-1* at 50 ng/μl alongside 20 ng/μl of the GFP template plasmid pHA#766 and the standard injection cocktail described in *Dickinson et al. (2015)*. Progeny from this injection were screened as described (*Dickinson et al., 2015*). Consistent with Miguel-Aliaga et al., we noticed that the tagged-SMN protein was expressed in all blastomeres throughout embryonic development with redistribution from the nucleus to the cytoplasm during mitotic stages (data not shown). The presence of GFP::SMN during such early stages indicates that GFP::SMN is maternally transmitted during germline development (*Miguel-Aliaga et al., 1999*).

## RNAi studies

RNAi studies involved animals from an RNAi-enhanced background (KP3948) (*Kennedy et al., 2004*), neuron-specific RNAi-sensitized background (TU3401) (*Calixto et al., 2010*), cholinergic neuron-specific RNAi-sensitized background (XE1581), or GABA neuron-specific RNAi-sensitized background (XE1375) (*Firnhaber and Hammarlund, 2013*). Aldicarb response, pumping rates, and RNA quantification were evaluated in animals that had been reared for at least two generations on HT115 bacteria containing control vector L4440, C41G7.1/*smn-1(RNAi)*, C26C6.2/*goa-1(RNAi)*, T06A10.1/ *mel-46(RNAi)*, or Y37D8A.23/*unc-25(RNAi)* (*Kamath and Ahringer, 2003*). Primer sequences used to generate the PCR products specific to each gene of interest can be found on the Kim Lab Stanford University website (http://cmgm.stanford.edu/~kimlab/primers.12-22-99.html).

## *C. elegans* behavioral assays

*Aldicarb resistance assay:* 1 mM aldicarb assays were completed in at least three independent trials blinded to genotype (n ≥ 30 animals/genotype) as described in previous work (*Mahoney et al., 2006*; *Sato et al., 2009*). Paralysis induced by aldicarb was scored as inability to move or pump in response to prodding with a platinum wire. Experiments involving *smn-1(ok355)*, *smn-1(rt248)* or *mel-46(tm1739)* animals were completed at the early L4 stage. All other aldicarb experiments were done with young adult animals. *Pharyngeal pumping:* Assays were performed blinded to genotype

as previously described (*Dimitriadi et al., 2010*). Pumping events were scored as grinder movement in any axis. Average pumping rates (± Standard Error of the Mean (SEM)) were pooled from at least two independent trials (n > 20 animals/genotype). Experiments involving *smn-1(ok355), smn-1(rt248)* or *mel-46(tm1739)* animals were completed at day three post-hatching (animals were kept at 25°C for two days and then 20°C for one day). Pumping experiments involving all other genotypes were done with young adult animals.

## *C. elegans* light level microscopy

Animals were mounted on 2% agar pads and immobilized using 30 mg/mL BDM (Sigma) in M9 buffer. *Dorsal cord protein localization:* Images were obtained as Z-stacks of the dorsal cord above the posterior gonad reflex (100x objective, Zeiss (Jena, Germany) AxioImager ApoTome and Axiovision software v4.8). For MEL-46::GFP analysis, a set area was defined for each image along the dorsal cord (25 μm x 5 μm). Using ImageJ (RRID:SCR_003070), a uniform threshold was used to eliminate background. The number (density), mean fluorescence (intensity) and area (size) for MEL-46::GFP granular structures were calculated using the ImageJ 'particle analyzer' program. For synaptic protein localization, mean puncta width (meanfixedwidth), intensity (meanfixedvolume) and linear density (fixedwidthlineardensity) were quantified with an in-house developed program called 'Punctaanalyser' using MatLab software (v6.5; Mathworks, Inc., Natick, MA, USA; RRID:SCR_001622) (*Kim et al., 2008*). At least three independent trials (n > 17 animals/genotype) were performed. For data sets involving *smn-1(ok355), smn-1(rt248),* or *mel-46(tm1739)* animals, all genotypes were examined at the early L4 stage, while other data sets were collected with young adult stage animals. *GFP Fluorescence Quantification:* GFP images of L4 animals were acquired (10x objective, Zeiss V20 stereoscope and Axiovision software v4.8). Mean GFP fluorescence was quantified using ImageJ (RRID:SCR_003070). A threshold was set to eliminate background fluorescence. For each data set, thresholds were kept constant. Average fluorescence values (±SEM) were combined from at least three independent trials for n > 25 animals/genotype; however certain backgrounds containing *rtEx855[mel-46(+)]* had a lower n (reported in legends) as these animals went sterile and/or did not throw many progeny carrying the *mel-46* array. Ratios in *Figure 4H* and *Figure 5A* were calculated as average mean fluorescence for each genotype in the *rtIs56* background and divided by their respective average mean fluorescence in the control *rtIs57* background. Ratio SEM was calculated by summing the SEM for each population (see *Figure 4—figure supplement 2D*). All representative images shown were analyzed as part of data collection.

## *C. elegans* total RNA isolation, cDNA synthesis and qPCR

For each RNA sample, animals were synchronized by collecting eggs for 6 hr from gravid adults on large seeded NGM plates. After two days at 25°C, young adult progeny were washed off, rinsed and flash frozen. Total RNA was extracted after a 15 min Trizol (Thermo Fisher, Waltham, Massachusetts) incubation. 1 ng total RNA was used for reverse transcription with either the miScript II RT kit (Qiagen #218160) for miRNA or the SuperScript III First-Strand Synthesis Supermix kit (Invitrogen #11752050) for mRNA. Methodology followed manufacturer's instructions. miRNA levels were determined in a 10 μl reaction using miScript SYBR Green PCR kit (#218073, Qiagen, Venlo, Netherlands) and 300 nM of mature miR-2 primer/probe. miR-60 was used to normalize miR-2 expression as it is not expressed in the nervous system where SMN-1 or MEL-46 were knocked-down. Forward primer sequences for miR-2 and miR-60 were, respectively: 5'-TATCACAGCCAGCTTTGATGTGC-3' and 5'-TATTTATGCACATTTTCTAGTTCA-3'. A universal reverse probe was provided by Qiagen. Primer sequences for *act-1*: 5'-acgccaacactgttcttttcc-3' and 5'-gatgatcttgatcttcatggttga-3' (*Ly et al., 2015*). Primer sequences for 18S rRNA: 5'-TTGCTGCGGTTAAAAAGCTC'3' and 5'-CCAACCTCAAACCAGCAAAT-3' (*Essers et al., 2015*). The stability of miR-60, 18S rRNA, and *act-1* housekeeping RNAs were evaluated using the 'model-based approach to estimation of expression variation' (*Andersen et al., 2004*). mRNA levels were determined in a 10 μl reaction using Power SYBR Green PCR Master Mix (Thermo Fisher Scientific # 4368706), and 300 nM of each primer. PGK-1 was used to normalize *gar-2* expression, as the mammalian orthologue has been used previously as a housekeeping gene for experiments involving SMN (*Abera et al., 2016*; *Simard et al., 2007*). Primer sequences for *gar-2*: 5'-CCTGAACTCTCATTGCCCTTTATTGATGC-3' and 5'-CTAGCAGTCCCTTGCATTGAAAC-3'. Primer sequences for *pgk-1*: 5'-GGCCCTCGACAACCCAGCTCGTC-3' and 5'-

CGGCGGAGGAATGGCCTATACC-3. All reactions were performed in triplicate. Melting curve analysis and electrophoresis in agarose gel of every PCR product was conducted after each qRT-PCR to control amplification specificity. Gene expression level was calculated as the fold change of relative DNA amount of a target gene in a target sample and a reference sample normalized to a reference gene using the comparative $\Delta\Delta$CT method as previously described (*Kurrasch et al., 2004*).

### Embryonic spinal MN culture, miR-128a quantification and Western blot

E13.5 mouse MNs were isolated from WT (FVB/NJ; RRID:IMSR_JAX:001800) and SMA mice (FVB, $Smn^{-/-};SMN2^{tg/0}$; generated by crossing lines RRID:IMSR_JAX:005058 and RRID:IMSR_JAX:005024) (*Riessland et al., 2010*) as described (*Wiese et al., 2001*). Isolated mouse MNs were differentiated 10 days in NB/B27 media supplemented with growth factors to promote survival; brain derived neurotrophic factor (BDNF) 10 ng/ml, ciliary neurotrophic factor (CNTF) 10 ng/ml and glial-derived neurotrophic factor (GDNF) 50 ng/ml. Fifty percent of medium was replaced every three days. To reduce the amount of glia and fibroblasts in culture, 1 μM cytosine arabinoside (AraC) was added at day 3.

After 10 days in vitro culture, total RNA was extracted from MNs using the mirVana total RNA isolation kit (Thermo Scientific). Nanodrop was used to measure RNA amount. Using 100 ng of total RNA, miR-128 expression levels were determined by real time PCR with mature miR-128a primer/probe (TaqMan MicroRNA Assays, #4427975, Thermo Scientific). Actin-beta was used to normalize miR-128a expression. Primer sequences for actin-beta: 5'-agccatgtacgtagccatcc-3' and reverse 5'-ctctcagctgtggtggtgaa-3'. Methodology followed manufacturer's instruction (*Kye et al., 2014*).

Proteins were extracted from motor neurons, after 10 days in vitro culture, using RIPA buffer and protease inhibitor cocktail (*Smith et al., 2014*). Expression of m2R and $\beta$-actin was measured using Western blot. Antibodies against m2R (ABCAM, ab109226; RRID:AB_10858602; 1:1000) and $\beta$-actin (Santa Cruz, sc-47778; RRID:AB_626632; 1:1000) were used to detect proteins. Methoctramine (Sigma, M105) was treated 48 hr from DIV3 to DIV5 in various concentrations. After 5 days of in vitro culture, neuronal morphology was visualized with Tau (Santa Cruz, A-10) staining. Axon length was analyzed with ImageJ (RRID:SCR_003070).

### Statistical analysis

Log-rank test, two-tailed Mann-Whitney $U$-test, or $t$-test were used for *C. elegans* statistical analysis. The Mann Whitney $U$-test was chosen over $t$-test for experiments where homogeneity could not be assured (i.e. RNAi; extrachromosomal arrays; or potential maternal loading from a heterozygous parent). $t$-test was used to determine significance for spinal motor neuron Western blot quantification and qPCR quantification.

## Acknowledgements

We thank the Streit lab for providing the PRM8[*mel-46(+)*] construct. We also thank the Jorgensen, Fire and Ahringer labs for sharing numerous plasmids. The Kaplan lab provided strains containing synaptic protein localization markers. Additional strains were provided by the CGC, funded by the NIH Office of Research Infrastructure Programs (P40 OD010440). This work was supported by the SMA Foundation and NIH NINDS (NS066888) to ACH, as well as NIH pre-doctoral training grants (2T32MH020068-11 and 5T32MH020068-12) and a NIH NINDS Ruth L Kirschstein National Research Service Award (F31NS089201) to PJO. The Kye lab was supported by the Deutsche Forschungsgemeinschaft (KY96/1-1), Cologne fortune and cure SMA foundation.

## Additional information

### Funding

| Funder | Grant reference number | Author |
|---|---|---|
| Spinal Muscular Atrophy Program Project | P01 NS066888 | Anne C Hart |
| National Institute of Neurological Disorders and Stroke | NS066888 | Anne C Hart |

| Deutsche Forschungsge-meinschaft | KY96/1-1 | Min Jeong Kye |
| Spinal Muscular Atrophy Foundation | SAH1516 | Min Jeong Kye |
| Universität zu Köln | 152/2014 | Min Jeong Kye |
| National Institute of Neurological Disorders and Stroke | F31NS089201 | Patrick J O'Hern |
| National Institute of Mental Health | 2T32MH020068-11 | Patrick J O'Hern |
| National Institute of Mental Health | 5T32MH020068-12 | Patrick J O'Hern |

The funders had no role in study design, data collection and interpretation, or the decision to submit the work for publication.

## Author contributions

PJO, Conceptualization, Data curation, Formal analysis, Funding acquisition, Investigation, Methodology, Writing—original draft, Writing—review and editing; ICG, JB, EJLS, JS, NC, Data curation, Formal analysis; DL, Supervision, Methodology, Writing—review and editing; MJK, Formal analysis, Supervision, Funding acquisition, Investigation, Methodology, Writing—original draft, Writing—review and editing; ACH, Conceptualization, Resources, Formal analysis, Supervision, Funding acquisition, Investigation, Methodology, Writing—original draft, Project administration, Writing—review and editing

## Author ORCIDs

Patrick J O'Hern, http://orcid.org/0000-0002-7633-2779
Anne C Hart, http://orcid.org/0000-0001-7239-4350

## Ethics

Animal experimentation: Invertebrate experimental procedures were carried out in accordance with the Brown University Institutional Animal Care and Use Committee (IACUC) guidelines, which adheres to the USDA: Animal Welfare Act & Regulations and the Guide for the Care and Use of Laboratory Animals. The Brown University IACUC does not require permit or protocol approval for experiments involving invertebrate animals models. All mammalian experimental procedures were performed according to the institutional animal care committee guidelines of University of Cologne. Additionally, animal care and all surgical procedures were performed according to the institutional animal care committee guidelines and the German animal welfare laws and approved under the reference numbers 84-02.05.20.13.042 and UniKoeln_Anziege§4.16.020 of the LANUV (Landesamt für Natur, Umwelt und Verbraucherschutz NRW) state agency of North-Rhine-Westphalia.

# Additional files

## Supplementary files

• Supplementary file 1. Summary of *C. elegans* experiments. (A) Summary of synaptic protein localization experiments; percent change from control for all metrics. (B) Summary of *C. elegans* phenotypes for selected loss of function alleles from this study as well as those identified in previous literature. *Related to Figures 1–6 and their figure supplements.*

• Supplementary file 2. *C. elegans* strains and other materials utilized. (A) Strains used for this work; related to materials and methods. (B) Primers and ssODNs used for this work; related to materials and methods. (C) Arrays/integrated lines/alleles with abbreviation in text and figures.

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
