## [Decision Letter]

Thank you for submitting your article "Decreased SMN causes deleterious increases in neuronal M2 muscarinic receptors due to microRNA misregulation" for consideration by *eLife*. Your article has been reviewed by two peer reviewers, and the evaluation has been overseen by a Reviewing Editor and a Senior Editor. The reviewers have opted to remain anonymous.

The reviewers have discussed the reviews with one another and the Reviewing Editor has drafted this decision to help you prepare a revised submission.

The authors harness the power of C.elegans genetics to interrogate the defects incurred by loss of SMN on neuromuscular function. Using a series of approaches, the authors implicate the worm ortholog of Gemin3 (*mel-46*), the miR-2 and the M2 muscarinic receptor. The study demonstrates the power of a genetically tractable system to understand fundamental biology. Based on their data and previously published work, the authors propose the following:

1) SMN-1 acts genetically through MEL-46/Gemin3. Molecularly, SMN-1 is necessary to achieve normal levels of MEL-46/Gemin3.

2) Normal levels of Gemin3 are necessary for correct miRNA processing/function, including *mir-2.*

3) *mir-2* is necessary to repress *gar-2*. GAR-2 is an M2 receptor that is predicted to inhibit synaptic release at cholinergic NMJs and thus needs to be kept at relatively low levels for optimal NMJ function

These conclusions are based on a number of relatively indirect measures, which are for the most part not sufficient to support all claims. In essence, a number of qPCR experiments are required to support a number of conclusions. Specifically:

1) The authors propose that reduction of SMN-1 results in lower levels of MEL-46. This is based on one genetic interaction: the fact that *mel-46* overexpression rescues part of the defects observed in *smn-1(lf)* animals, most prominently, it suppresses the aldicarb resistance of *smn-1(lf)* animals. However, it does not rescue the pharyngeal defect of *smn-1(lf)*, suggesting some kind of tissue specificity. Since the connection to miRNA processing/function is given by MEL-46/Gemin3, I think it's important to provide a more direct link and the authors need to measure *mel-46* (RNA using qPCR and/or protein) in *smn-1(lf)*.

This is important given that an alternative explanation is not satisfactorily rejected. Specifically, the authors discard an effect of MEL-46 in stabilizing the maternal contribution of SMN-1 by quantifying GFP fluorescence from a very nice SMN-1::GFP allele generated with Cas9, over whole worms, in wt animals with or without MEL-46 overexpression. However, given that in all phenotypic assays the effect of MEL-46 is only seen in the *smn-1(lf)* background but not in the wt background, it is not clear if one would expect to see any changes in SMN-1 abundance in wt animals. A more direct test would have been to look at the remaining GFP in the *smn-1(lf)* progeny from heterozygous mothers and see whether this changes with overexpression of MEL-46. Also, given the tissue specificity observed phenotypically, it may be more revealing to focus e.g. on motorneurons. Although if levels are too low to see GFP, a western blot against GFP may already provide some information.

2) The connection of SMN to miRNAs through Gemin3 is solely based on previous publications reporting the dysregulation of certain miRNAs when SMN or Gemin3 are decreased. There is no direct connection in this work between this pathway and *mir-2*, the miRNA the authors find to be important for NMJ function. Based on the data provided, *mir-2* could be acting in the NMJs in a parallel pathway that has no connection to SMN-1 or MEL-46. To strengthen this link, the authors need to measure *mir-2* levels in *smn-1(lf), mel-46* mutants or overexpression strains. However, one has to wonder whether there is a connection at all: The authors claim to provide a link (although very indirect) with an epistasis analysis shown in Figure 4 where they look at a fluorescent reporter for *gar-2*, the predicted target of *mir-2*. The authors compare a reporter with the wt 3'UTR and one without the mir-2 binding site in wt animals or *smn-1(lf)*. In wt animals, the fluorescence ratio between the two reporters is ~1 and in *smn-1(lf)* animals it goes up by 5% (to 1.05). The authors suggest this is consistent with *mir-2* levels being lower in *smn-1(lf)* (because of lower mel-46/Gemin3, which is not shown) and that leads to *gar-2* derepression. In addition to this being an extremely small effect that would need additional data to be convincing (e.g. qPCR of *gar-2* mRNA in N2 v miR-2 mutant animals), there is a problem which is that the increase in ratio is not as one would expect from the reporter with the wt 3'UTR being derepressed (numerator going up) in *smn-1(lf)* but rather the reporter with the mutant 3'UTR is decreased in *smn-1(lf)* (denominator going down) (Figure 4—figure supplement 2, panels C and D). There is no other connection provided between the effect that *smn-1* and *mel-46* may have on the NMJ and whatever function *mir-2* might have there as well.

3) The phenotypic and reporter analyses are consistent with *gar-2* being a target of *mir-2*. Given the small magnitude of repression observed, this point could be made more convincingly, for example by mutating the *mir-2* binding site in the 3'UTR of *gar-2*, which should not be too difficult given that the authors are able to generate alleles using Cas9.

[Editors' note: further revisions were requested prior to acceptance, as described below.]

Thank you for resubmitting your work entitled "Decreased microRNA levels lead to deleterious increases in neuronal M2 muscarinic receptors in Spinal Muscular Atrophy models" for further consideration at *eLife*. Your revised article has been favorably evaluated by a Senior Editor, a Reviewing Editor, and two reviewers.

The manuscript has been improved but there are some remaining issues that need to be addressed before acceptance, as outlined below:

The authors have done a highly appreciated effort to incorporate the reviewers’ comments and suggestions and I think this has paid off in strengthening their model. However, one concern remains about a critical link in the story: The changes in *mir2* measurements (Figure 5), an important and good addition, needs to be done with better controls. Currently, the authors report *mir-2* levels relative to *mir-60* levels. This is another miRNA that might also be affected by the manipulations done in this experiment. It is not clear at all from looking at the raw data provided whether in these experiments *mir-2* is decreasing or *mir-60* is increasing, in fact the latter seems more supported by the data. There is also a problem of reproducibility in this experiment, with variances of >10 Ct between replicates. It is strongly suggested that given the importance of this experiment, it should be repeated with additional normalization controls, which should include at least one non-miRNA RNA (U6, U18, 5S RNA). One suggestion for improving reproducibility is to use a remarkably well-working protocol published by Kien Ly, Suzanne J. Reid, Russell G. Snell as "Rapid RNA analysis of individual *Caenorhabditis elegans*" in MethodsX 2 (2015) 59-63.

---

## [Author Response]

*[…] These conclusions are based on a number of relatively indirect measures, which are for the most part not sufficient to support all claims. In essence, a number of qPCR experiments are required to support a number of conclusions. Specifically:*

*1) The authors propose that reduction of SMN-1 results in lower levels of MEL-46. This is based on one genetic interaction: the fact that mel-46 overexpression rescues part of the defects observed in smn-1(lf) animals, most prominently, it suppresses the aldicarb resistance of smn-1(lf) animals. However, it does not rescue the pharyngeal defect of smn-1(lf), suggesting some kind of tissue specificity. Since the connection to miRNA processing/function is given by MEL-46/Gemin3, I think it's important to provide a more direct link and the authors need to measure mel-46 (RNA using qPCR and/or protein) in smn-1(lf).*

*This is important given that an alternative explanation is not satisfactorily rejected. Specifically, the authors discard an effect of MEL-46 in stabilizing the maternal contribution of SMN-1 by quantifying GFP fluorescence from a very nice SMN-1::GFP allele generated with Cas9, over whole worms, in wt animals with or without MEL-46 overexpression. However, given that in all phenotypic assays the effect of MEL-46 is only seen in the smn-1(lf) background but not in the wt background, it is not clear if one would expect to see any changes in SMN-1 abundance in wt animals. A more direct test would have been to look at the remaining GFP in the smn-1(lf) progeny from heterozygous mothers and see whether this changes with overexpression of MEL-46. Also, given the tissue specificity observed phenotypically, it may be more revealing to focus e.g. on motorneurons. Although if levels are too low to see GFP, a western blot against GFP may already provide some information.*

We appreciate the reviewer’s criticism and the opportunity to show a more direct connection between SMN-1 and MEL-46. To confirm that loss of SMN-1 affects the function and/or levels of MEL-46, we have now generated two independent integrated lines expressing MEL-46 tagged with GFP behind the control of the cholinergic-specific *unc-17* promoter (Figure 2). Using these, we were able to evaluate the effects of SMN-1 loss on MEL-46 levels and localization in cholinergic neurons. This seemed the most specific approach to quantify the effect of SMN-1 loss on MEL-46 in motor neurons. After crossing both independent integrated lines into the *smn-1(ok355*) background, we found an obvious defect in MEL-46 localization in dorsal cord motor neurons processes. We find that MEL-46::GFP in wild type control animals is sparsely localized to punctate structures along the dorsal cord (see Figure 2). These structures are likely similar to axonal granules containing MEL-46, which have been reported in mammalian tissue culture studies (Todd et al., 2010a; Todd et al., 2010b; Zhang et al., 2006). In *smn-1(ok355*) animals, MEL-46::GFP is found in many more granules, with an average decrease in MEL-46::GFP fluorescence within granules (Figure 2-Figure 1). MEL-46::GFP mislocalization and decreased intra-granule levels in *smn-1(ok355)* cholinergic neurons support our model in which MEL-46 function is perturbed when SMN-1 levels decrease.

Unfortunately, we were unable to directly evaluate levels of maternally-loaded SMN-1::GFP in *smn-1(ok355)* homozygous animals. The simplest strategy would be to tag the endogenous *smn-1* gene on the balancer chromosome used to propagate *smn-1(ok355).* Using the CRISPR-based approach outlined in the manuscript and suggested by the reviewer, we attempted to GFP-tag endogenous *smn-1* on balancer chromosomes *hT2* and *ccIs425*. We were unsuccessful after multiple attempts. Therefore, we cannot unequivocally conclude that increased MEL-46 does not lead to increased stabilized maternally-loaded SMN-1. We make this caveat clear in the revised text (see “changes made to text” below). Given the severe MEL-46 localization defects we observe in *smn-1(ok355)* animals, however, it is more likely that increased MEL-46 proteins levels ameliorate *smn-1(ok355)* defectsby increasing MEL-46 within critical granules in motor neurons, like those in the dorsal cord. See Figure 2 and Figure 2—figure supplement 1.

Changes made to text for Editor Point 1:

“MEL-46 might act together with or downstream of SMN-1 in pathways necessary for NMJ function. […] Our findings suggest that SMN-1 impairs MEL-46 function, which could contribute *to smn-1(ok355)* synaptic defects (Dimitriadi et al., 2016).”

“Using the same CRISPR-based method, we were unable to tag endogenous *smn-1* on balancer chromosomes necessary for maintaining the *smn-1(ok355)* line. […] Therefore, although we cannot rule out stability of maternally-loaded SMN-1 as a contributing factor, the large effect that decreased SMN-1 has on MEL-46 localization favors a mechanism in which MEL-46 overexpression rescues *smn-1(ok355)* defects, at least in part, by restoring MEL-46 functional deficits in this background.”

*2) The connection of SMN to miRNAs through Gemin3 is solely based on previous publications reporting the dysregulation of certain miRNAs when SMN or Gemin3 are decreased. There is no direct connection in this work between this pathway and mir-2, the miRNA the authors find to be important for NMJ function. Based on the data provided, mir-2 could be acting in the NMJs in a parallel pathway that has no connection to SMN-1 or MEL-46. To strengthen this link, the authors need to measure mir-2 levels in smn-1(lf), mel-46 mutants or overexpression strains. However, one has to wonder whether there is a connection at all: The authors claim to provide a link (although very indirect) with an epistasis analysis shown in Figure 4 where they look at a fluorescent reporter for gar-2, the predicted target of mir-2. The authors compare a reporter with the wt 3'UTR and one without the mir-2 binding site in wt animals or smn-1(lf). In wt animals, the fluorescence ratio between the two reporters is ~1 and in smn-1(lf) animals it goes up by 5% (to 1.05). The authors suggest this is consistent with mir-2 levels being lower in smn-1(lf) (because of lower mel-46/Gemin3, which is not shown) and that leads to gar-2 derepression. In addition to this being an extremely small effect that would need additional data to be convincing (e.g. qPCR of gar-2 mRNA in N2 v miR-2 mutant animals), there is a problem which is that the increase in ratio is not as one would expect from the reporter with the wt 3'UTR being derepressed (numerator going up) in smn-1(lf) but rather the reporter with the mutant 3'UTR is decreased in smn-1(lf) (denominator going down) (Figure 4—figure supplement 2, panels C and D). There is no other connection provided between the effect that smn-1 and mel-46 may have on the NMJ and whatever function mir-2 might have there as well.*

We thank the reviewers for these suggestions. The work we have consequently done with qPCR analysis of miR-2 and *gar-2* transcripts greatly strengthens the manuscript. We focused specifically on how decreasing SMN-1 or MEL-46 within neurons impacts mature miR-2 levels and *gar-2* mRNA levels. To do this, we undertook RNAi studies using animals sensitive to RNAi by feeding only in neurons (Calixto et al., 2010). These animals were placed on bacteria expressing SMN-1 or MEL-46 RNAi constructs for at least two generations. We found that knock-down of either SMN-1 or MEL-46 resulted in a large (~80%) decrease in miR-2 levels. These results are consistent not only with a role for both SMN-1 and MEL-46 in miR-2 biogenesis, but further suggest a relationship between SMN-1 and MEL-46 in a pathway important for proper miRNA function. In measuring *gar-2* transcript from these samples, we found that this decrease in miR-2 levels did not result in significantly increased *gar-2* mRNA levels. This was unsurprising as *gar-2* mRNA levels were also not significantly increased in *mir-2(gk259)* loss of function animals compared to N2 control animals (Figure 4). Previous reports have found that changes in miRNA levels do not necessarily result in changes in target transcript levels. Instead, miRNAs may bind their targets and inhibit translation without transcript destabilization (Cloonan, 2015; Selbach et al., 2008). Overall, these data support our model and further suggest that SMN-1 and MEL-46 not only impact miRNA activity, but may play a role in miRNA biogenesis.

See Figure 4 and Figure 5.

Changes made to text for Editor Point 2:

“We also assessed the effect of miR-2 loss on *gar-2* transcript levels and found no significant difference in *gar-2* mRNA levels between wild type animals and *mir-2(gk259)* loss of function animals (Figure 4). […] Previous studies have reported that miRNAs can influence protein synthesis of targets without destabilizing mRNA levels (Cloonan, 2015; Selbach et al., 2008).”

“Increased GAR-2 translation in animals lacking SMN-1 might be due to decreased mature miR-2 levels. […] These results suggest neuronal miR-2 levels are decreased when MEL-46 or SMN-1 levels decrease.”

3) The phenotypic and reporter analyses are consistent with gar-2 being a target of mir-2. Given the small magnitude of repression observed, this point could be made more convincingly, for example by mutating the mir-2 binding site in the 3'UTR of gar-2, which should not be too difficult given that the authors are able to generate alleles using Cas9.

We agree. CRIPSR/Cas-9-based genome editing was used to scramble the microRNA binding site in the *gar-2* 3’UTR. Animals with a scrambled endogenous miR-2 binding site in the *gar-2* 3’UTR were resistant to aldicarb, when compared to control animals (Figure 4). Furthermore, animals with the scrambled binding site in the *gar-2* 3’UTR had increased levels of *gar-2* messenger RNA (Figure 4). These results strengthen the model in which *gar-2* is a direct target of miR-2. This result is perplexing, however, since *mir-2(gk259)* animals did not have significantly increased *gar-2* transcript levels. We note that other miR-2 family members likely bind this site, including miR-43, miR-250, and miR-797. Additionally, our qPCR results come from whole animal lysate. Thus, the increase in *gar-2* messenger RNA likely results from the inability of all miR-2 family members to bind the *gar-2* 3’UTR throughout all tissues. These family members may act differently than miR-2, resulting in transcript destabilization when bound. Furthermore, since miR-2 appears to be limited to the nervous system, the effect it has on *gar-2* transcript in neurons may be minimized by levels of the more broadly-expressed *gar-2* transcript. This may explain why a significant difference in transcript levels is seen when the binding site is scrambled, but not in *mir-2(gk259)* loss of function animals.

See Figure 4 and Figure 4.

Changes made to text for Editor Point 3:

“To determine if miR-2 regulates GAR-2 expression directly, we examined the consequences of perturbing the putative miR-2 binding site in the *gar-2* 3’UTR. […] Disruption of the 3’UTR site likely inhibits binding of other miR-2 family members, likely contributing to the effect we observe (Ibanez-Ventoso et al., 2008).”

[Editors' note: further revisions were requested prior to acceptance, as described below.]

*The manuscript has been improved but there are some remaining issues that need to be addressed before acceptance, as outlined below:*

*The authors have done a highly appreciated effort to incorporate the reviewers’ comments and suggestions and I think this has paid off in strengthening their model. However, one concern remains about a critical link in the story: The changes in mir2 measurements (Figure 5), an important and good addition, needs to be done with better controls. Currently, the authors report mir-2 levels relative to mir-60 levels. This is another miRNA that might also be affected by the manipulations done in this experiment. It is not clear at all from looking at the raw data provided whether in these experiments mir-2 is decreasing or mir-60 is increasing, in fact the latter seems more supported by the data. There is also a problem of reproducibility in this experiment, with variances of >10 Ct between replicates. It is strongly suggested that given the importance of this experiment, it should be repeated with additional normalization controls, which should include at least one non-miRNA RNA (U6, U18, 5S RNA). One suggestion for improving reproducibility is to use a remarkably well-working protocol published by Kien Ly, Suzanne J. Reid, Russell G. Snell as "Rapid RNA analysis of individual Caenorhabditis elegans" in MethodsX 2 (2015) 59-63.*

We thank the reviewer for these suggestions. We undertook the additional experiments suggested by the reviewer. We increased the number of samples from 3 to 6 for each group and examined miR-2 expression levels relative to 3 different RNAs: a microRNA (miR-60), a ribosomal RNA (18S), and a messenger RNA *(act-1).* After either *mel-46* and *smn-1* RNAi treatments, we observed decreased miR-2 levels, normalized to any of these 3 RNAs relative to empty RNAi (Figure 5; Figure 5—figure supplement 1). In the revised manuscript, we report miR-2 levels relative to miR-60 levels in the main figures and include miR-2 levels relative to *act-1* and 18S rRNA in the supplementary figures (Martinez et al., 2008). We utilized previously reported amplification primers for both 18S rRNA (Essers et al., 2015) and *act-1* (Ly et al., 2015). This extended analysis makes clear that miR-2 levels decrease when *smn-1* or *mel-46* function decreases.

Choosing an appropriate control RNA can be challenging; here we examined 3 different possibilities. We evaluated the stability of each of the putative “control RNAs” under the different experimental conditions using the “model-based approach to estimation of expression variation” (Andersen et al., 2004). This mathematical model is used to describe the “stability value” of candidate genes- taking in account variability within a group of samples under the same treatment conditions and between groups of samples under different treatment conditions. Lower “stability values” for a given group of genes indicate more stable expression. We find that miR-60 is the most stable RNA (stability value = 0.250), compared to 18S rRNA or *act-1* (0.316 and 0.376, respectively). As miR-60 is the most stable candidate, we use it as a control in the main text figures.

See Figure 5 and Figure 5—figure supplement 1.

Changes made to text for Editor Point 1:

“After neuron-specific RNAi knock-down of either SMN-1 or MEL-46, we found decreases in mature miR-2 levels (Figure 5; Figure 5—figure supplement 1), but no change in *gar-2* transcript levels (Figure 5).”

“Primer sequences for *act-1*: 5’-acgccaacactgttctttcc-3’ and 5’-gatgatcttgatcttcatggttga-3’ (Ly et al., 2015). Primer sequences for 18S rRNA: 5’-TTGCTGCGGTTAAAAAGCTC’3’ and 5’-CCAACCTCAAACCAGCAAAT-3’ (Essers et al., 2015). The stability of miR-60, 18S rRNA, and *act-1* housekeeping RNAswere evaluated using the ‘model-based approach to estimation of expression variation’ (Andersen et al., 2004).”

“miR-2 levels were decreased in neurons when either SMN-1 or MEL-46 were decreased. Quantification of mature miR-2 for empty*(RNAi), smn-1(RNAi)*, and *mel-46(RNAi)* young adult animals relative to housekeeping miRNA miR-60. *t-*test, two-tailed (n=6 for each condition).”

“(C) miR-2 levels were decreased in neurons when either SMN-1 or MEL-46 were decreased. Quantification of mature miR-2 for empty*(RNAi), smn-1(RNAi)*, and *mel-46(RNAi)* young adult animals relative to housekeeping gene *act-1*. […] Quantification of mature miR-2 for empty*(RNAi), smn-1(RNAi)*, and *mel-46(RNAi)* young adult animals relative to housekeeping rRNA 18s. *t-*test, two-tailed (n=6 for each condition).”